**REPORT**

# Mitochondrial dysfunction compromises ciliary homeostasis in astrocytes

Olesia Ignatenko[1], Satu Malinen[1], Sofiia Rybas[1], Helena Vihinen[2], Joni Nikkanen[3], Aleksander Kononov[4], Eija S. Jokitalo[2], Gulayse Ince-Dunn[1], and Anu Suomalainen[1,5]

**Astrocytes, often considered as secondary responders to neurodegeneration, are emerging as primary drivers of brain disease. Here we show that mitochondrial DNA depletion in astrocytes affects their primary cilium, the signaling organelle of a cell. The progressive oxidative phosphorylation deficiency in astrocytes induces FOXJ1 and RFX transcription factors, known as master regulators of motile ciliogenesis. Consequently, a robust gene expression program involving motile cilia components and multiciliated cell differentiation factors are induced. While the affected astrocytes still retain a single cilium, these organelles elongate and become remarkably distorted. The data suggest that chronic activation of the mitochondrial integrated stress response (ISRmt) in astrocytes drives anabolic metabolism and promotes ciliary elongation. Collectively, our evidence indicates that an active signaling axis involving mitochondria and primary cilia exists and that ciliary signaling is part of ISRmt in astrocytes. We propose that metabolic ciliopathy is a novel pathomechanism for mitochondria-related neurodegenerative diseases.**

## Introduction

Astrocytes are essential cells of the central nervous system (CNS), maintaining ionic balance, blood–brain barrier integrity, synapse function, and metabolic homeostasis (Sofroniew and Vinters, 2010; Wang and Bordey, 2008). When stressed by disease or injury, astrocytes transition into a state termed reactive astrogliosis, characterized by functional, morphological, and molecular remodeling that can be protective or toxic (Anderson et al., 2016; Bush et al., 1999; Liddelow et al., 2017; Yun et al., 2018). Despite the essential functions of astrocytes in CNS metabolism, the molecular programs induced by metabolic stresses remain insufficiently understood.

Pathogenic variants in genes encoding mitochondrial proteins result in rare inherited neurodegenerative diseases (Gorman et al., 2016) and secondary mitochondrial dysfunctions contribute to common neurological diseases (Lin and Beal, 2006; Wang et al., 2019). Recent data show that mitochondrial dysfunction in astrocytes is sufficient to induce severe brain pathology in mice (Ignatenko et al., 2018; Ignatenko et al., 2020; Murru et al., 2019). Our studies have shown that loss of the mitochondrial DNA-helicase Twinkle (Twnk) in astrocytes (TwKO^astro; KO for knockout) and consequent progressive mitochondrial DNA (mtDNA) loss

lead to pervasive reactive astrogliosis with progressive vacuolation of brain parenchyma, reactive microgliosis, myelin disorganization, and premature death (Ignatenko et al., 2018). Interestingly, the brain pathology of TwKO^astro mice closely resembles human spongiotic encephalopathies caused by mtDNA depletion (Kollberg et al., 2006; Palin et al., 2012). In contrast, Twnk deletion and mtDNA loss in cortical and hippocampal excitatory neurons causes few signs of disease until manifesting acute-onset neurodegeneration at 7 mo of age (Ignatenko et al., 2018). The brains of TwKO^astro mice undergo metabolic reprogramming with induction of mitochondrial integrated stress response (ISRmt), while mice with neuronal loss of Twnk show no signs of ISRmt (Ignatenko et al., 2020). These data indicate that CNS responses to mitochondrial stress are cell-type specific.

Here we report that mtDNA loss in astrocytes causes anomalous induction of a motile ciliogenesis program, mediated by transcription factors (TFs), RFX (Regulatory Factor X) family and FOXJ1 (Forkhead Box J1), and major lipid metabolic remodeling. Our findings open an exciting new front for astrocyte research, relevant for both primary mitochondrial and neurological diseases with secondary mitochondrial dysfunction.

[1]Stem Cells and Metabolism Research Program, Faculty of Medicine, University of Helsinki, Helsinki, Finland; [2]Institute of Biotechnology, University of Helsinki, Helsinki, Finland; [3]Cardiovascular Research Institute, University of California, San Francisco, CA; [4]Cancer Research UK, University of Manchester, Manchester, UK; [5]HUS Diagnostics, Helsinki University Hospital, Helsinki, Finland.

Correspondence to Anu Suomalainen: anu.wartiovaara@helsinki.fi; @AWartiovaara;  Gulayse Ince-Dunn gulayse.dunn@helsinki.fi; @GulayseDunn.



## Results and discussion

### Astrocytes lacking Twinkle helicase develop mtDNA depletion

To investigate astrocyte-specific metabolic responses to mitochondrial dysfunction, we purified cortical astrocytes from 3- to 3.5-mo-old TwKO[astro] and control mice, using magnetic beads coated with antibodies against astrocyte-specific surface antigen ATP1B1 (ACSA-2; Batiuk et al., 2017) and carried out bulk RNAseq analysis. Astrocytes were enriched in the ACSA-2+ fraction (Fig. 1 A). Endothelial cell marker Cspg4 was also enriched, as previously observed (Batiuk et al., 2017). At this age, TwKO[astro] mice show early-stage disease with mild gliosis and sparse vacuoles in brain parenchyma (Fig. S1). Myelin disorganization, neuronal loss, and mtDNA depletion in total cortical lysates are not yet observed (Fig. 1 B; Ignatenko et al., 2018). Unsupervised principal component analysis of RNAseq data showed that control and TwKO[astro] samples clustered separately (Fig. 1 C). Expression levels of 1,131 genes were upregulated and 408 were downregulated (|log2(FC)| > 0.3 and q-value <0.1; Fig. 1 D). Astrocytes from TwKO[astro] mice showed a profound mtDNA depletion and mtDNA-encoded transcripts (Fig. 1, B, D, and E).

### Astrocytes lacking Twinkle induce a partial ISRmt and molecular markers of reactive astrogliosis

A close examination of cellular stress response genes revealed induction of several defining components of ISRmt in astrocytes purified from TwKO[astro]: activating TFs (Atf3, Atf5) and Trib3 kinase that govern the response; mitochondrial folate metabolism genes (Mthfd2, Mthfd1l); Asns that catalyses conversion of aspartate to asparagine, and the metabokine Gdf15 (Fig. 1, D and F). This response showed disease-stage and tissue specificity lacking several components of ISRmt detected in total brain lysates from advanced TwKO[astro] disease-stage (Psat1, Phgdh, Cth, Cbs [Ignatenko et al., 2020]), and from ISRmt of skeletal muscle (Fgf21 [Forsström et al., 2019]).

We then examined previously published datasets of reactive astrogliosis markers associated with different brain pathologies (Escartin et al., 2021; Liddelow et al., 2017; Zamanian et al., 2012; Table S1). Out of a total of 56 such marker genes, expression of 22 was changed in TwKO[astro] (Fig. 1 G). Genes encoding intermediate filaments and factors involved in immune responses (e.g., Nfatc2 and Nfatc4) were robustly upregulated (Fig. 1 G). These data show that mitochondrial dysfunction-related reactive astrogliosis signature overlaps only partially with those of other CNS stresses.

### Mitochondrial dysfunction in astrocytes alters brain lipid metabolism

Gene ontology analysis of downregulated transcripts in TwKO[astro] purified astrocytes indicated major changes in lipid metabolic pathways (Fig. 1 H). Analysis of lipids and lipid-like molecules in an untargeted metabolomics dataset (Ignatenko et al., 2020) from TwKO[astro] cortical lysates showed progressive impairment of lipid homeostasis (Fig. 1 I). At the early 2.3-mo-old timepoint, only two ceramides and a cholesteryl ester were depleted while at the in 3.2-mo-old time point, 106 out of 172 identified lipid metabolites were changed (Fig. 1 I and Table

S2). Prominent changes were reflected also in lipid storage: ultrastructural analysis and biochemical detection with BODIPY493/503 probe revealed an accumulation of lipid droplets in TwKO[astro] mice at a late disease stage (5–8 mo of age; Fig. 1, J and K). Collectively, our data demonstrate that mtDNA depletion in astrocytes induces a severe, progressive defect in brain lipid metabolism.

Cytoplasmic lipid accumulation in astrocytes could result from mitochondrial beta-oxidation impairment. In this case, shuttling lipids to lipid droplets could serve as a protective mechanism against lipotoxicity (Liu et al., 2015; Nguyen et al., 2017). Recent reports propose that astrocytes import and metabolize lipids generated by neurons, which may further increase the lipid load in astrocytes with defective beta-oxidation (Ioannou et al., 2019; Liu et al., 2015; Liu et al., 2017).

### Mitochondrial dysfunction in astrocytes induces a ciliogenic program through RFX and FOXJ1 transcription factors

Intriguingly, the top five upregulated pathways in TwKO[astro] astrocytes were related to cilia (Fig. 2 A). The majority of differentiated eukaryotic cells, including astrocytes, possess a primary signaling cilium specialized to sense and integrate external signals critical for cell proliferation and differentiation (Dahl, 1963; Karlsson, 1966; Kasahara et al., 2014; Sterpka and Chen, 2018). Surprisingly, the ciliary pathways upregulated in TwKO[astro] astrocytes were related to motile cilia (Fig. 2 A) that facilitate liquid movement at the surface of lumen-facing cells and are not present in astrocytes. To explore the ciliary components induced in the TwKO[astro] dataset, we used SysCilia and CiliaCarta databases (van Dam et al., 2013; van Dam et al., 2019). TwKO[astro] astrocytes showed a remarkable upregulation of 64 out of 280 SysCilia and 163 of 791 CiliaCarta genes, while only 4 and 13 genes, respectively, were downregulated (Fig. 2 B and Fig. S2 A and Table S3).

Next, we asked whether specific TF consensus sequences were enriched in the promoters of genes upregulated in astrocytes from TwKO[astro] mice. The analysis revealed an enrichment of motifs recognized by RFX TF family (Fig. 2 C and Fig. S2 B). RFX1-4 TFs are master regulators of ciliogenesis in a wide range of cell types (Santos and Reiter, 2008; Thomas et al., 2010). Analysis of publicly available RFX1-3 ChIP-Seq datasets derived from mouse multiciliated ependymal cells (Lemeille et al., 2020), revealed that RFX-binding sites were significantly overrepresented in genomic regions proximal to transcription start sites of genes upregulated in TwKO[astro] astrocytes compared to negative gene sets (Fig. 2 D and Table S4). Rfx1 transcript level was slightly increased, while other RFX family member mRNAs were unchanged (Fig. 2 E and Fig. S2 C). These findings suggest RFX-family transactivation or their recruitment to promoters in our astrocyte population. Furthermore, Trp73 and Foxj1, two TFs that control motile ciliogenesis and differentiation of multiciliated cells, were also upregulated (Fig. 2 F and Fig. S2 C). Trp73-encoded TP73 triggers a cascade of multiciliated cell differentiation (Napoli and Flores, 2016; Nemajerova et al., 2016), key components of which (Cdkn1a, Myb, Foxj1, Ank3, and Six3) are all upregulated in our dataset (Fig. 2 F). In summary, the transcriptional response to

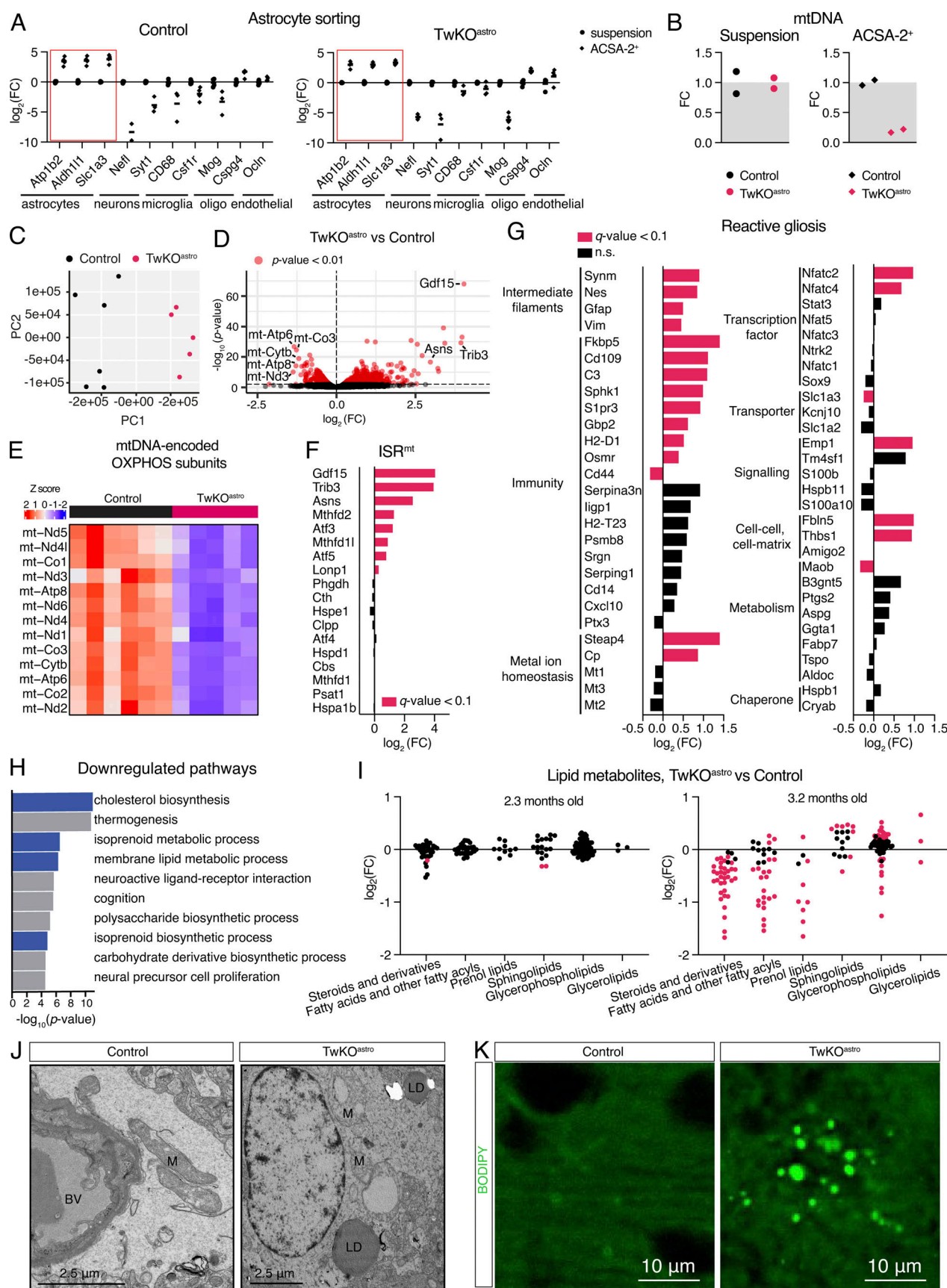

Figure 1. **mtDNA depletion in astrocytes leads to cellular stress responses and alters brain lipid metabolism. (A)** Enrichment of cell-specific genes in ACSA-2+ fraction compared to unsorted cortical cell suspension, measured using RT-qPCR. Genes of interest are normalized to Hmbs gene expression level. **(B)**

MtDNA amount, normalized to nuclear DNA (single copy gene Rbm15). Measured using qPCR. **(C)** Transcriptome of astrocytes, principal component (PC) analysis. **(D)** Transcriptome of astrocytes, all identified genes. Gene symbols in the volcano plot denote those most downregulated genes that are encoded in the mtDNA and those most upregulated genes that are ISRmt components (red dots indicate P value <0.01). **(E)** Heatmap of mtDNA transcripts. **(F)** Transcripts encoding ISRmt components (gene list is curated based on Forsström et al., 2019). **(G)** Transcripts that mark reactive astrogliosis (gene list is curated based on Escartin et al., 2021; Liddelow et al., 2017; Zamanian et al., 2012). See also Table S1. **(H)** Gene ontology pathway enrichment analysis of downregulated genes. Blue color denotes pathways related to lipid metabolism. **(I)** Lipids and lipid-like molecules, metabolomics (dataset from Ignatenko et al., 2020). Pink dots indicate P value <0.01 at 2.3 mo timepoint and q-value <0.1 at 3.2 mo timepoint. See also Table S2. FC = fold change. **(J)** Transmission electron microscopy, mouse brain cortex, 5–8-mo-old mice, images are representative of four mice per genotype. LD, lipid droplet; M, mitochondria. **(K)** BODIPY493/503 neutral lipid dye staining, 5-mo-old mice, cortex, images are representative of five mice per genotype. (C–H): RNA sequencing, astrocytes purified from Ctrl (*n* = 6) and TwKO^astro (*n* = 5) mouse brain cortical preparations, 3–3.5-mo-old mice. See also Table S6.

mitochondrial dysfunction in astrocytes converged on a robustly activated motile ciliogenic program.

FOXJ1 is a master regulator of motile ciliogenesis that works in coordination with RFX TFs and in the brain is presumed to be active exclusively in ependymal cells, the only multiciliated cell type in the CNS (Jacquet et al., 2009; Stubbs et al., 2008; Yu et al., 2008). Foxj1 expression in purified ACSA-2 fraction was markedly higher than in unsorted brain suspension both in control and TwKO^astro mice, suggesting that Foxj1 is normally expressed in adult astrocytes (Fig. 2, F and G). However, also ependymal cells may be captured by ACSA-2 preps as reported previously (Ohlig et al., 2021). To investigate the spatial Foxj1 expression, we used RNA-fluorescence in situ hybridization and found markedly increased expression of Foxj1 in TwKO^astro cerebral cortex parenchyma at 3.5 mo of age (Fig. 2 H and Fig. S2 D). Foxj1-positive puncta were present also in the cortical parenchyma of control mice indicating that expression of Foxj1 is not restricted to ependymal cells in the brain (Fig. 2 H; and Fig. S2, D and E). As expected, ependymal cells also showed a strong Foxj1 signal (Fig. 2 H). Our evidence suggests that Foxj1 can be expressed in astrocytes of the cerebral cortex of adult mice and is induced in response to mtDNA loss.

### Mitochondrial dysfunction in astrocytes induces expression of motile cilia components

To investigate the extent of motile cilia program induction in TwKO^astro, we curated a catalogue of genes that encode proteins that are either unique to motile cilia, unique to primary cilia or are pan-ciliary (Table S3). Axonemes of both primary and motile cilia comprise nine outer doublet microtubules, while motile cilia also harbor a set of distinct components essential for their movement (Fig. 3 A). The expression of genes that compose all key structures specific to motile cilia was induced in TwKO^astro astrocytes (Fig. 3 B). These included dynein arms and the cytoplasmic factors required for their assembly; the nexin–dynein regulatory complex (nexin link) that is located between microtubule doublets; the central pair (an additional microtubule doublet); the radial spokes that protrude toward the central pair; and other factors involved in the assembly and motility of the axoneme (Fig. 3 B). The expression of most genes which are specific to primary cilia or are pan-ciliary was unchanged (Fig. 3 C). Proteomics analysis of purified TwKO^astro astrocytes also showed the induction of motile cilia factors: out of 11 detected proteins of motile cilia, 6 were upregulated (Fig. 3 D). Remarkably, out of 10 detected pan-ciliary proteins, 5 were induced at the protein but not RNA level (Fig. 3 D). These five proteins are

components of pan-ciliary intraflagellar transport (IFT) complex, essential for cilia formation and maintenance and implicated in the control of ciliary length (Broekhuis et al., 2013; Avasthi and Marshall, 2012; Ishikawa and Marshall, 2017). This evidence suggests that transcriptional induction of motile ciliogenesis program in astrocytes with mitochondrial dysfunction results in stabilization of axonemal proteins.

Together these data show that mtDNA loss in adult mouse brain induces Foxj1 expression and a motile ciliogenic program in astrocytes, a cell type that normally possesses a single primary cilium. Intriguingly, ectopic expression of FOXJ1 in *Xenopus* and *Danio rerio* is sufficient to induce formation of functionally motile cilia in cells normally devoid of these organelles (Stubbs et al., 2008; Yu et al., 2008).

### Motile ciliogenesis program is activated by metabolic and growth factor–related astrocyte stress

The abnormal motile ciliogenesis and multiciliated cell differentiation program induction in TwKO^astro astrocytes prompted us to explore in previously published datasets whether such a response is detected also in other astrocyte stress models (Anderson et al., 2016; Guttenplan et al., 2020; Li et al., 2019), ageing astrocytes (Boisvert et al., 2018), or other mouse models with mitochondrial dysfunction (Kühl et al., 2017). In TwKO^astro astrocytes, 62 out of 92 genes encoding motile cilia components or ciliogenic and multiciliated cell differentiation factors were changed compared to controls (Fig. 3 E and Table S3). In contrast, five mouse models with heart-specific knockouts of mitochondrial gene expression factors, including Twnk, did not show such changes (Fig. 3 E), indicating that even TwKO responses are cell-type specific. Cultured astrocytes inhibited for EGF signaling (EGFR silencing or HBEGF withdrawal) displayed a robust upregulation of the motile ciliogenesis program, including induction of Foxj1 (Fig. 3 E; Li et al., 2019). In contrast, reactive astrocytes stimulated with pro-inflammatory IL1+TNFα+C1q cocktail or sorted from mice with spinal cord injury displayed partial downregulation of the motile ciliogenesis program (Fig. 3 E; Guttenplan et al., 2020). Finally, astrocytes sorted from aged mouse brain showed no changes (Fig. 3 E; Boisvert et al., 2018). These data suggest that mitochondria–cilia communication axis is cell-type specific, even in response to mitochondrial dysfunction.

### Astrocytic cilia are elongated and contorted in TwKO^astro mice

The robust upregulation of the motile multiciliary program in TwKO^astro astrocytes prompted us to image astrocyte cilia in

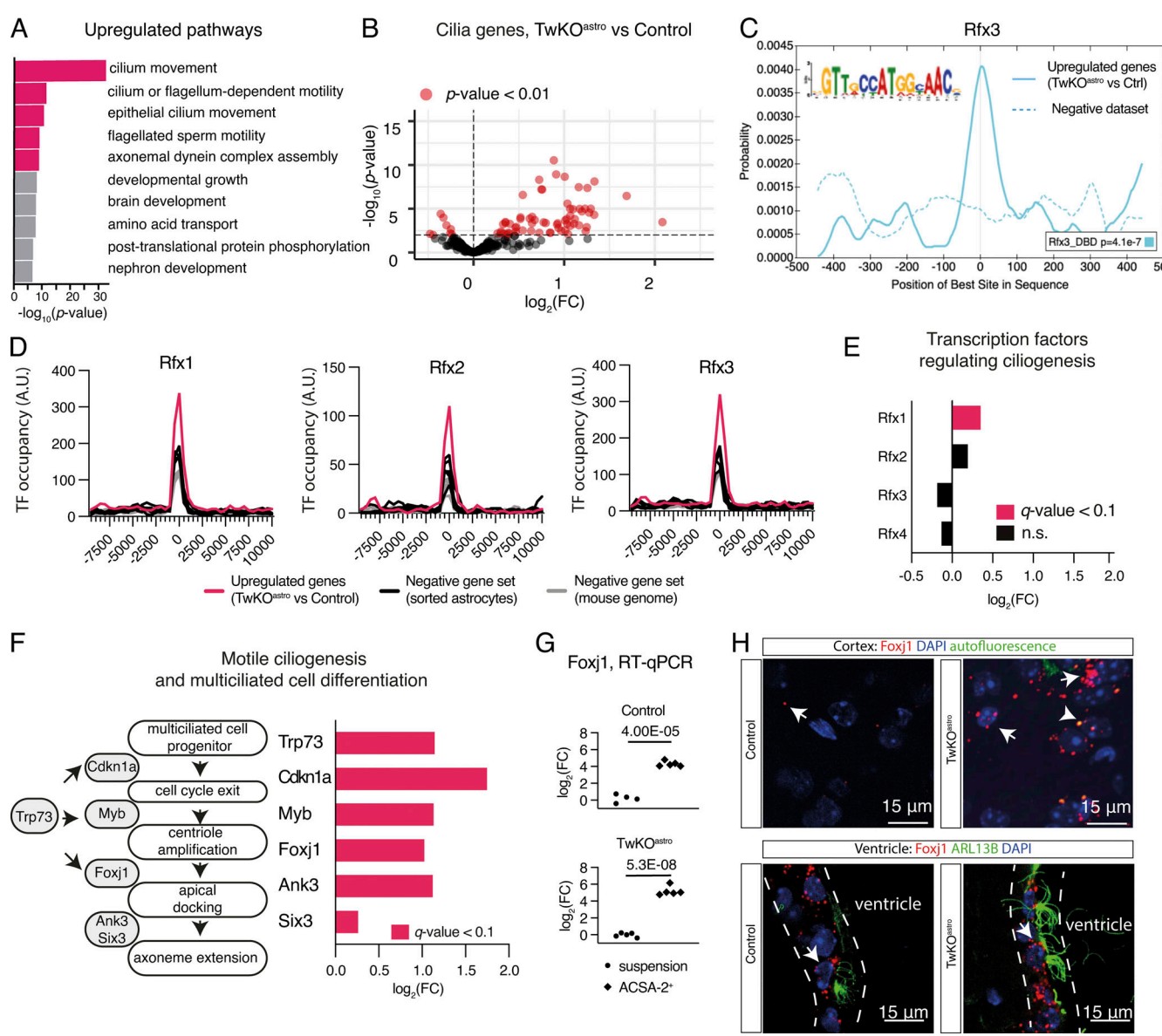

Figure 2. **MtDNA depletion in astrocytes purified from TwKO^astro mice induces motile ciliogenesis program. (A–F)** RNA sequencing, astrocytes purified from TwKO^astro compared to control mice. The dataset is described in Fig. 1. **(A)** Gene ontology pathway enrichment analysis of upregulated genes in astrocytes of TwKO^astro mice. Pathways involved in cilia motility are marked in red. **(B)** Genes expressing ciliary proteins in astrocytes of TwKO^astro mice (SysCilia Gold Standard gene list [van Dam et al., 2019]). See also Fig. S2 A and Table S3. **(C)** Motif enrichment analysis of promoters of upregulated genes in TwKO^astro astrocytes reveals RFX consensus motif (RFX3 motif enrichment is shown here, see also Fig. S2 B). **(D)** Distribution graphs of the TF-binding sites proximal to the promoters of upregulated genes (ChIP-seq data from Lemeille et al., 2020). X-axis represents distance from TSS (transcription starting site), which equals point zero. Randomly selected gene sets either from the whole mouse genome or genes expressed in astrocytes, but not changed in our dataset, constitute negative gene sets. See also Table S4. **(E)** Transcription factors that regulate ciliogenesis in the brain; mRNA levels in TwKO^astro astrocytes. **(F)** Key steps of multiciliated cell differentiation, from cell cycle exit to motile cilia formation; schematic representation (left). The expression of the key regulators of cilia biogenesis in astrocytes of TwKO^astro mice. **(G)** Foxj1 expression in purified astrocytes (ACSA-2+ fraction) compared to unsorted cell suspension; RT-qPCR, normalized to Ywhaz transcript level. Symbols represent individual preparations ($n$ = 5 per genotype). P values calculated using unpaired two-tailed parametric $t$ test. **(H)** Foxj1 expression; RNA-fluorescence in situ hybridization, 3.5-mo-old mice, images are representative of 3-4 mice per genotype. Dashed lines indicate the ependymal cell layer. Puncta specific for Foxj1 channel were analyzed (arrows). Puncta that overlap with autofluorescence or ARL13B signal (arrowhead) were excluded as non-specific signal. Mice also express AAV-gfaABC1D-Arl13b-eGFP to visualize cilia. See also Fig. S2, D and E.

TwKO^astro mice. Even at 4.5–5 mo of age, in the mice with advanced disease (Fig. S1; Ignatenko et al., 2018), the astrocytes were monociliated (ciliary axoneme protein ARL13B co-stained with GFAP or pan-astrocytic marker ALDH1L1; Fig. 4, A and B). Astrocytes positive for either marker revealed a shift in length distribution toward longer cilia in TwKO^astro mice: 2.0–6.5 μm in controls and 2.5–8.2 μm in TwKO^astro cortex (Fig. 4 C). Such elongation is consistent with induced protein levels of IFT components (Fig. 3 D; Broekhuis et al., 2013; Avasthi and Marshall, 2012; Ishikawa and Marshall, 2017). To analyze cilia shape, we classified them as straight, bent, or contorted (Fig. 4 D). Contorted cilia in TwKO^astro included S-shaped and

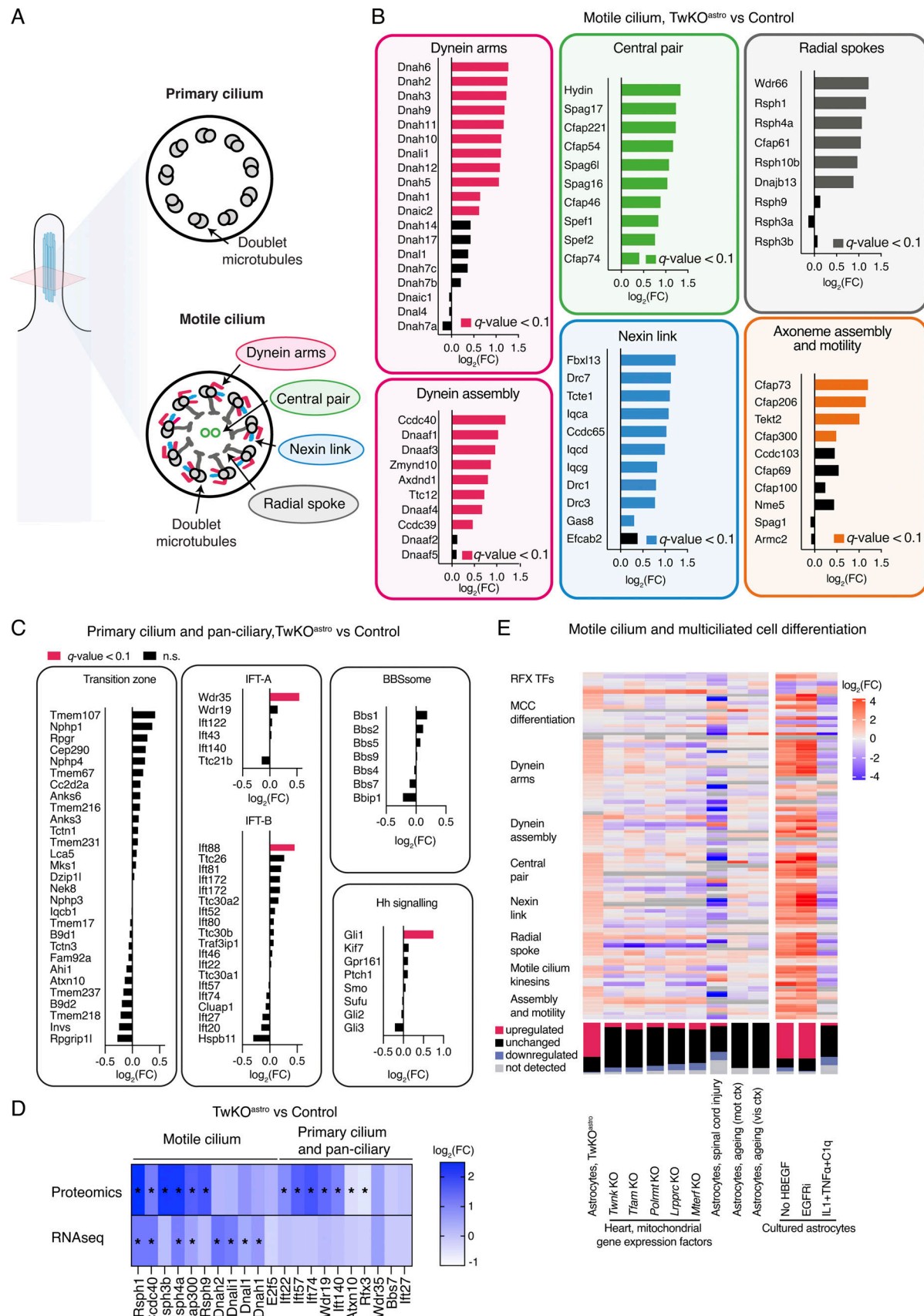

Figure 3. **Motile cilia components are induced in astrocytes with mitochondrial dysfunction or growth signaling inhibition. (A–E)** Includes RNA sequencing, astrocytes purified from TwKO[astro] compared to control mice. The dataset is described in Fig. 1. **(A)** Schematic representation of cilia cross-section.

Axonemes of both primary and motile cilia comprise nine doublet microtubules. Motile cilia also harbor components that are not present in primary cilia. **(B)** Expression of structural components specific to motile cilia and other factors involved in axoneme assembly and motility in astrocytes of TwKO^astro mice compared to control mice. **(C)** Expression of pan-ciliary and primary cilia factors in astrocytes purified from TwKO^astro compared to control mice. IFT = intraflagellar transport; BBBsome = a component of the basal body; Hh signaling = hedgehog signaling pathway. **(D)** Protein and RNA expression of motile or pan-ciliary and specific to primary cilia factors in astrocytes purified from TwKO^astro compared to control mice. Note that IFT22, IFT57, IFT74, WDR19, and IFT140 are IFT complex components (Table S3). Stars indicate q-value <0.01. **(E)** Regulation of motile ciliogenic program in astrocytes upon various insults and upon tissue-specific mitochondrial dysfunction. Heat maps (top panel) and stacked bar charts (bottom panel). Datasets are from this study (described in Fig. 1) and from Anderson et al. (2016); Boisvert et al. (2018); Guttenplan et al. (2020); Kühl et al. (2017); Li et al. (2019). MCC, multiciliated cell; SCI, spinal cord injury; mot ctx, motor cortex; vis ctx, visual cortex. Expression data for some of these genes are also presented in Figs. 2, E and F, S2 C, and 3, B and D.

corkscrew-like morphologies, and long cilia occasionally appeared to form several loops (Fig. 4 E, bottom panel). Such extreme morphologies never occurred in control mice. Generally, the proportion of cilia with contorted morphology was higher in TwKO^astro than in controls (Fig. 4 F). Collectively, our results indicate that astrocytic mitochondrial dysfunction remarkably modifies the structure of the primary cilium.

Further, we investigated cellular localization of induced motile cilia factors. A component of nexin-dynein regulatory complex GAS8 showed strong cytoplasmic induction in cortical astrocytes but no detectable trafficking into cilia (Fig. 4, G and H). Conversely, despite RSPH4A induction at the protein level (Fig. 3 D), we did not detect any immunofluorescence signal in the cortex of TwKO^astro mice (Fig. S2, F and G). These data suggest defective trafficking of some of the induced motile cilia components into the cilium and consequent accumulation in the cytoplasm.

### Ependymal multiciliated cells show no major abnormality upon mitochondrial dysfunction

Next, we asked whether multiciliated ependymal cells in TwKO^astro mice were affected. Initially, using the Rosa26-CAG-LSL-tdTomato reporter mice, we determined that GFAP-73.12-Cre driver line used to generate TwKO^astro mice was active in ependymal cells (Fig. 5 A). These cells in 5-mo-old TwKO^astro mice showed a marked cytochrome c oxidase deficiency (the respiratory chain complex IV, partially encoded by mtDNA), whereas the activity of nuclear-encoded complex II was preserved, similar to the findings in cortical astrocytes of TwKO^astro mice (Fig. 5 B; Ignatenko et al., 2018). Anti-ARL13B immunostaining and scanning electron microscopy indicated that ependymal cells were multiciliated in TwKO^astro mice (Fig. 5, C and D). However, we did not test the functionality of the ependymal motile cilia. Together, our data show that both astrocytes and ependymal cells survive mitochondrial gene expression stress, and only parenchymal astrocytes show considerable ciliary abnormality.

In conclusion, we show that mitochondrial dysfunction in astrocytes induces remarkable structural aberration of primary cilia. Astrocytes with mtDNA loss induce an anomalous expression of a motile cilia program governed by TFs FOXJ1 and the RFX family. The signaling axis between mitochondria and cilia has been recently suggested in studies of dividing cultured cells, after toxin-mediated or genetic mitochondrial respiratory defect (Bae et al., 2019; Burkhalter et al., 2019; Failler et al., 2020). Our evidence indicates that mitochondrial respiratory chain

dysfunction leads to a rewiring of the ciliary pathway in the differentiated CNS astrocytes in vivo.

Despite the multiciliary program induction, TwKO astrocytes possess a single cilium. The astrocytes show a shift in length distribution toward long, sometimes extremely contorted primary cilia. Whether the activation of a motile ciliogenesis program represents a physiological and functional shift from immotile to motile cilia, or is a pathological state associated with aberrant expression and cytoplasmic accumulation of motile cilia factors (such as GAS8) remains unknown. The mechanisms may involve: (1) axoneme elongation caused by induction of ciliogenic factors and stabilization of intraflagellar transport components; (2) bending of elongated axonemes due to physical constraints in the tight extracellular space; (3) aberrantly induced motile cilia components accumulating in the cytoplasm or being trafficked into primary cilia, modifying axonemal structure. Recent studies established that ciliary morphology is changed in a variety of brain pathologies, including common neurodegenerative disorders (Ki et al., 2021; Tereshko et al., 2022). While the functional significance of such changes remains still insufficiently understood, they may alter the distribution of ciliary membrane receptors and signaling properties.

Traditionally, cilia were classified as primary or motile based on motility and/or axonemal structure (9 + 0 and 9 + 2 microtubular arrangement, respectively). Most differentiated cell types, including astrocytes, are thought to possess a single immotile primary 9 + 0 cilium lacking the motile cilia components. However, in vivo experimental evidence for this is lacking for a majority of cell types (Matsumoto et al., 2019). Intriguingly, non-canonical arrangements of microtubular organization of primary cilia (Odor and Blandau, 1985; Gilroy et al., 1995; Sun et al., 2019; Kiesel et al., 2020 *Preprint*) and primary cilia expressing motile cilia components (Cho et al., 2021 *Preprint*) were reported. These studies emphasize that cilium types are diverse, some with hybrid motile and primary properties. Our study agrees with such conclusions and indicates that motile cilia program can be induced in differentiated cells also as a result of disease.

The potential signals to provoke motile cilia program include metabolic components of ISRmt, remodeling critical metabolites of anabolic metabolism, such as amino acids, TCA cycle intermediates, nucleotides related to ISRmt (Nikkanen et al., 2016; Ignatenko et al., 2020; Forsström et al., 2019), as well as lipids. Here, we present metabolic ciliopathy as a new contributor to pathophysiology of primary mitochondrial brain diseases and as a relevant candidate to contribute in neurodegenerative pathologies associated with secondary mitochondrial dysfunction.

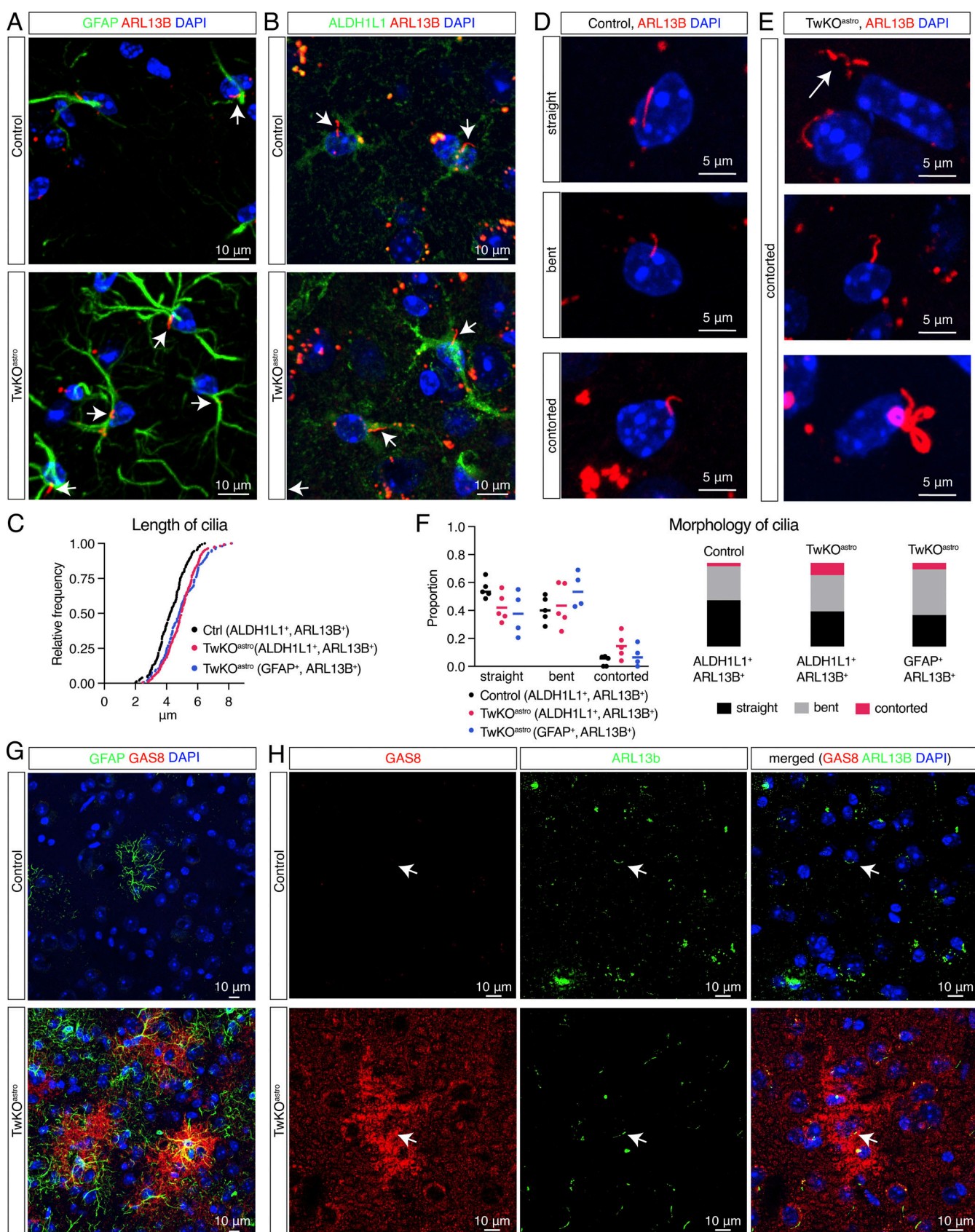

Figure 4. **Astrocyte cilia elongate and become contorted upon astrocytic mitochondrial dysfunction. (A–H)** Cerebral cortex, 4.5–5-mo-old mice. **(A and B)** Ciliary marker ARL13B immunostaining in GFAP+ (A) or ALDH1L1+ (B) astrocytes in control and TwKO^astro mice. **(C)** Length distribution of cilia in TwKO^astro and control mice. ARL13B signal in ALDH1L1+ or GFAP+ cells analyzed. $n$ = 5 mice per genotype, 6 fields of view per mouse. Total number of cilia: control

ALDH1L1+, ARL13B+ = 148; TwKO<sup>astro</sup> ALDH1L1+, ARL13B+ = 174; TwKO<sup>astro</sup> GFAP+, ARL13B+ = 75; control GFAP+, ARL13B + cells were too few for analysis. Control vs. TwKO<sup>astro</sup> (ALDH1L1+ARL13B+): Kolmogorov–Smirnov test P value <0.0001, D = 0.5703; one-way analysis of variance: Pr (>F) = 6.76e−09. Dots represent individual cilia. **(D)** Representative images of cilia morphology in control mice. Immunostaining against ARL13B. **(E)** Cilia are contorted and elongated in TwKO<sup>astro</sup> mice, immunostaining against ARL13B. **(F)** Quantification of cilia morphology in TwKO<sup>astro</sup> and control astrocytes, based on the dataset from C. Dots represent an average per mouse. Stacked bar graphs show average per genotype. **(G and H)** Motile cilia component GAS8 co-immunostained with GFAP (G) or ARL13B (H). Arrows point to ARL13B-positive cilia.

## Limitations of the study

We report that astrocytes, a cell type that normally possesses a primary immotile cilium, induce a motile cilia program at RNA and protein level in response to mitochondrial dysfunction. Complementing this analysis with imaging approaches in vivo to observe the cilium structure is challenged by the difficulty to capture these small singular organelles in 3D space (see review history of this article). We investigated cilia in vivo using high-resolution confocal imaging using both antibodies and AAV-mediated expression of fluorescently labeled ciliary proteins. To resolve axonemal structure and positioning in the tissue context, we developed a pipeline for 3D correlative light-electron microscopy (3D-CLEM), but the resolution was insufficient. In cultured astrocytes imaged with electron transmission tomography, organelles were horizontal and internalized similar to other cultured cells (Kukic et al., 2016), not reflecting their position in the brain. Recent advances in FIB-SEM may be useful for cilia imaging in vivo, and pose a new exciting front in cilia biology studies (Sheu et al., 2022).

## Materials and methods

### Animal experimentation

Animal experiments were approved by The National Animal Experiment Review Board and Regional State Administrative Agency for Southern Finland, following the European Union Directive. Mice were maintained in a vivarium with 12-h light: dark cycle at 22°C and allowed access to food and water ad libitum. TwKO<sup>astro</sup> mouse lines (Ignatenko et al., 2018; Gfap73.12Cre+; Twnkloxp/loxp mice) were generated by crossing mice carrying floxed Twnk alleles (Twnkloxp/loxp or Twnk+/loxp) with Gfap73.12-Cre+ mice (JAX: 012886) creating a deletion of exons 2 and 3 of Twnk. Littermates were used as controls. Twnkloxp/+, Twnkloxp/loxp, Gfap73.12Cre+; Twnkloxp/+, Gfap73.12Cre+; Twnk+/+ were used as controls. The methodology for generating Twnkloxp/loxp mice is previously described (Nikkanen et al., 2016). These mice carried Y508C mutation in the targeted Twnk gene (Nikkanen et al., 2016). Using the same method as in Nikkanen et al. (2016), for this study we regenerated Twnkloxp/loxp mice without the Y508C mutation. Gfap73.12Cre+; Twnkloxp/loxp and Gfap73.12-Cre+; TwnkY508C/Y508C mice were indistinguishable in mtDNA brain pathology and were used interchangeably as TwKO<sup>astro</sup>. Reporter mice for Cre recombinase activity were generated using Ai14 tdTomato mice (JAX: 007914). Mice on C57Bl/6OlaHsd or mixed genetic background mice were used. For all experiments, mice were either terminally anesthetized by intraperitoneal injection of pentobarbital or were euthanized with $CO_2$.

### Astrocyte sorting from adult mice

Astrocytes were purified using magnetic beads coated with the ACSA-2 antibody (130-097-678; Miltenyi Biotec) according to manufacturer's instructions, with modifications (Batiuk et al., 2017; Holt et al., 2019). Briefly, cerebral cortex samples from four mice per sorting preparation were dissected to ice-cold PBS, minced with a scalpel, and transferred to C tubes (130-093-237; Miltenyi Biotec) with 3.6 ml of Buffer Z; 100 µl of Enzyme P, 40 µl Buffer Y, and 20 µl Enzyme A was added (Adult Brain Dissociation [P] Kit, #130-107-677; Miltenyi Biotec). The tissue was dissociated at 37°C upon continuous mechanical dissociation (program 37C_ABDK_01, Octodissociator, 130-095-937; Miltenyi Biotec) and pelleted by centrifugation (all centrifugations were 300 g, 10 min). Samples were placed on ice, and ice-cold PBS-BSA (0.5% BSA, A4503 or A4161; Sigma-Aldrich) was added to 10 ml, tissue was gently triturated using a serological pipette, passed through 70 µm filter, and centrifuged. To deplete debris, pellet was resuspended in 1.4 ml of PBS-BSA and 600 µl of debris removal solution, 2 ml of PBS-BSA was gently added on top, and centrifuged. To deplete myelin debris, pellets were resuspended in 330 µl of PBS-BSA and 40 µl of myelin Removal Beads II (#130-096-731; Miltenyi Biotec), incubated on ice for 15 min, after which PBS-BSA was added to 4 ml, and centrifuged. Pellets were resuspended, applied on pre-wet with PBS-BSA LS columns (130-042-401; Miltenyi Biotec) on a magnetic separator (130-091-051; Miltenyi Biotec), washed with PBS-BSA, followed by a wash with PBS-BSA. Flow-through was collected and centrifuged. To enrich astrocyte fraction, pellet was resuspended in 60 µl of PBS-BSA, incubated with blocking solution for 10 min on ice and with ACSA-2 antibodies conjugated to magnetic beads for 15 min on ice (MicroBead Kit, #130-097-679; Miltenyi Biotec). PBS-BSA was added to 2 ml and centrifuged. Pellet was resuspended in 1 ml of PBS-BSA, applied to two pre-wet with PBS-BSA MS columns consequently (130-042-201; Miltenyi Biotec) on a magnetic cell separator (130-042-108; Miltenyi Biotec). Columns were washed three times with 500 µl of PBS-BSA, after which 1 ml of PBS-BSA was added, column was removed from the magnetic stand, and liquid was pushed with a column plunger to an Eppendorf. The cells were centrifuged down, supernatant was discarded, and cell pellets were flash frozen in liquid nitrogen.

### Brain collection for histological analyses

Mice were transcardially perfused with ice-cold PBS followed by perfusion with ice-cold 4% formaldehyde solution in PBS. The brains were postfixed in 4% formaldehyde solution in PBS overnight at 4°C. For immunofluorescence, brains were stored in PBS with 0.02% sodium azide at 4°C and then for several days incubated in 30% sucrose in PBS solution before freezing. For

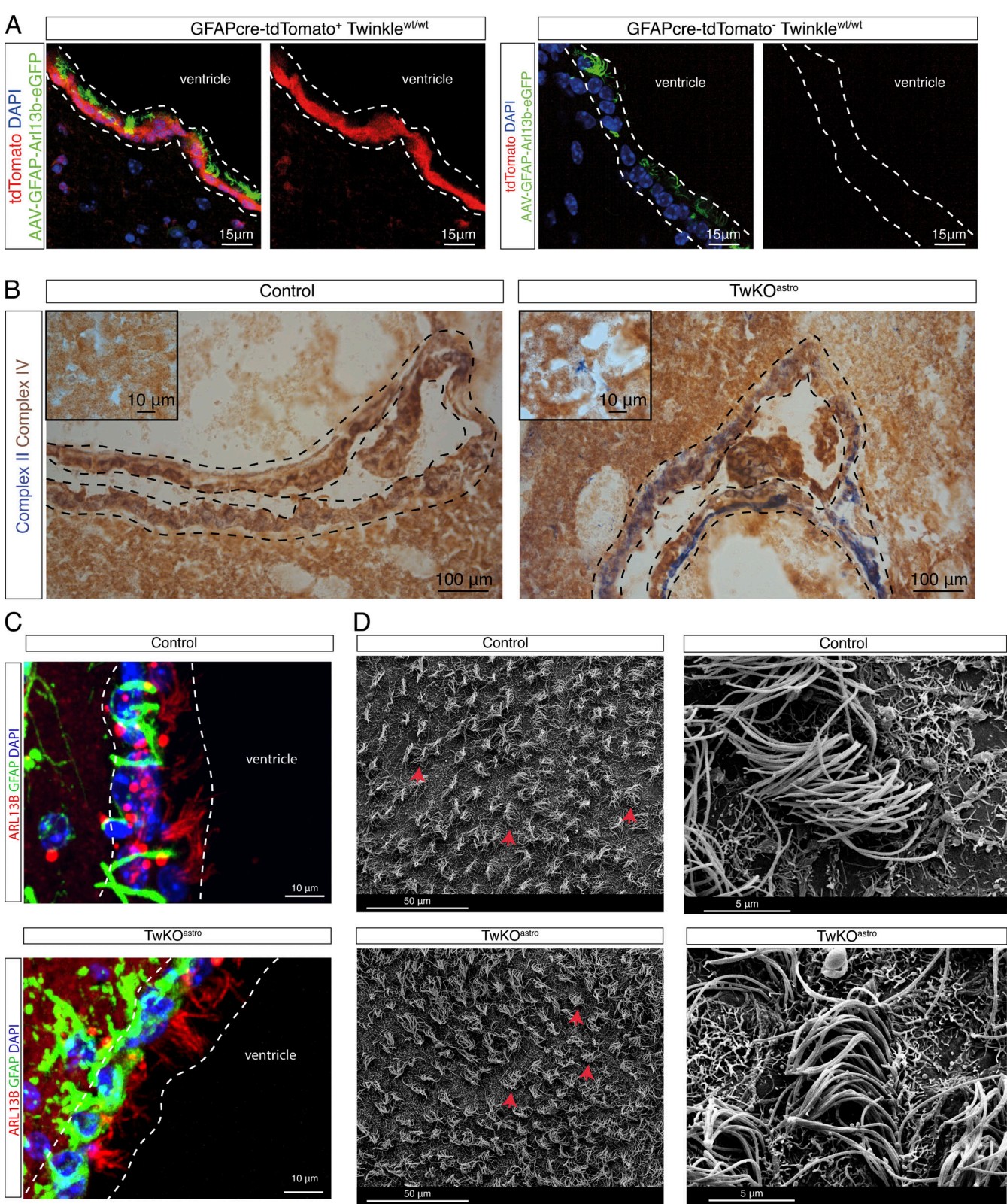

Figure 5.    **Ependymal multiciliated cells show no major abnormality upon mitochondrial dysfunction. (A)** GFAP73.12-Cre expression, ventricular lining, mouse brain. Expression of tdTomato is controlled by a floxed stop cassette. Dashed lines indicate the ependymal cell layer. Mice also express AAV-gfaABC1D-Arl13b-eGFP to visualize cilia. **(B)** Histochemical in situ enzymatic activity assay of Complex IV (brown precipitate) and Complex II (blue) of OXPHOS; frozen brain sections. Complex II subunits are encoded in nuclear DNA and Complex IV is partially encoded by mtDNA. 5.5-mo-old mice, images are representative of five mice per genotype. Ventricle, dashed lines, indicate ependymal cell layer. Insets are from the cortex. **(C)** Ependymal cell layer (dashed lines); ependymal cells and astrocytes (GFAP+) and motile cilia (ARL13B+), images are representative of five mice per genotype. **(D)** Scanning electron microscopy of the ventricle. Tufts of cilia (red arrows) on the surface of ependymal cells; images are representative of three mice per genotype.

RNA-fluorescence in situ hybridization, after postfixing brains were transferred to 30% sucrose in PBS for several days at 4°C, and then frozen in embedding media (O.C.T. compound #4583; Tissue-Tek) at –20°C. The brain sections were cut using Thermo Fisher Scientific Cryostar NX70 cryostat at 12–20 μm onto adhesive microscope slides (#TOM-11; Matsunami Glass) or Thermo Fisher Scientific (J1800BMNZ).

## Immunofluorescence

Frozen mouse brain sections were postfixed in 4% PFA at room temperature for 15–30 min. For heat induced epitope retrieval, the sections were incubated in 10 mM citric buffer, pH = 6 for 30 min at 80°C in pressure cooker, cooled down and incubated in 10% horse serum, 0.1% Triton X-100 (or 1.0% Tween20) in PBS for 1 h at room temperature. Samples were then incubated overnight at +4°C with primary antibodies diluted in the same solution or in an antibody diluent (#S3022; Agilent) and washed in 0.1% Triton X-100 or 1.0% Tween20 in PBS for 30–60 min, followed by incubation for 1 h at room temperature with secondary antibodies conjugated with Alexa Fluor fluorescent probes (Thermo Fisher Scientific) diluted in 10% horse serum, 0.1% Triton X-100 or 1.0% Tween20 in PBS. The samples were mounted with DAPI-containing mounting medium (#H-1200-10; Vectashield). Images were acquired using the Andor Dragonfly spinning disk confocal microscope using 60× Plan Apo VC objective with numerical aperture 1.2; 40× Apo LWD objective with numerical aperture 1.15 using water as imaging medium; at room temperature; camera Andor Zyla 4.2 sCMOS; imaging software Fusion 2.0. Maximum intensity projection images are presented in figure panels. Images were also acquired with a Zeiss Axio Imager epifluorescent microscope. For all images, only linear adjustments were applied. For all confocal images, maximum intensity projections are shown.

Antibodies used in the study Primary antibodies: rabbit anti-GFAP (1:500; AB5804; Sigma-Aldrich); mouse anti-ARL13B (1:500; #AB136648; Abcam); rabbit anti-ALDH1L1 (1:100; ab97117; Abcam); rabbit anti-GAS8 (1:100; #HPA041311; Atlas Antibodies); and rabbit anti-RSPH4A (1:100; #HPA031198; Atlas Antibodies). Secondary antibodies: chicken anti-mouse Alexa fluor 594 (1:500; A21201; Thermo Fisher Scientific); chicken anti-rabbit Alexa fluor 594 (1:500; A21442; Thermo Fisher Scientific); donkey anti-mouse Alexa fluor 488 (1:500; A21202; Thermo Fisher Scientific); and donkey anti rabbit Alexa fluor 594 (1:500; ab150068; Abcam).

## RNA-fluorescence in situ hybridization

Frozen mouse brain sections were dehydrated with ethanol and hybridized with a probe against mouse Foxj1 according to the manufacturer's instructions with modifications (#317091, #323110; ACD Systems and #FP1487001KT, #FP1488001KT; PerkinElmer). Briefly, sections were postfixed in 4% PFA in PBS for 30 min, rinsed twice in PBS, and dehydrated (5 min 50% EtOH, 5 min 70% EtOH, 2 × 5 min 100% EtOH), and dried at room temperature (5–10 min). A barrier was drawn with a PAP pen (ab2601; Abcam). Tissue sections were treated with 20–30 μl per section of Protease IV for 20 min at RT, rinsed in PBS, and incubated in the hybridization oven (40°C, HybEZ Hybridization

System, ACDbio) with probes and signal amplification reagents (20–30 μl of reagent per section; probe 120 min 40°C, 2 × 2 min washes in wash buffer RT, AMP1 30 min 40°C, 2 × 2 min washes in wash buffer RT, AMP2 15 min 40°C, 2 × 2 min washes in wash buffer RT, AMP3 30 min 40°C, 2 × 2 min washes in wash buffer RT, AMP4 15 min 40°C, 2 × 2 min washes in wash buffer RT; 20–30 μl of each reagent per section was used). Sections were mounted using antifade mounting medium with DAPI (H-1200-10; Vector Laboratories). Images were acquired using Andor Dragonfly spinning disk confocal microscope, 4 μm stacks with a 1 μm step were acquired. Signal was manually quantified from maximum intensity projection images using Fiji software by an investigator blind to genotypes. Clusters of a minimum of three puncta were quantified. The signal was quantified from three fields of view per mouse. Maximum intensity projections of confocal images are shown.

## Lipid staining

To visualize lipid droplets on mouse brain sections, BODIPY493/503 dye (intrinsic affinity to neutral and non-polar lipids) was used according to manufacturer's instructions (#D3922; Thermo Fisher Scientific). Briefly, mice were transcardially perfused with ice-cold PBS, followed by perfusion with ice-cold 4% formaldehyde solution. Brains were postfixed overnight, incubated for several days in 30% sucrose, frozen in the embedding media, and sectioned. Sections were incubated for 10 min in PBS, followed by 10–60 min in BODIPY solution (1:100-1:1,000 of the stock solution in PBS), washed twice for 5 min in PBS, mounted, and imaged.

## Analysis of cilia morphology

Cilia morphology was analyzed on mouse brain sections using anti-ARL13B and anti-ALDH1l1 antibodies as described above. ARL13B + signal in ALDH1L1+ cells was analyzed. Images were acquired with Andor Dragonfly spinning disk confocal microscope (7 μm stacks with a 0.5 μm step), and cilia morphology was analyzed using Imaris 9.5.1 software with Surfaces module. Settings for surfaces were adjusted to automatically detect cilia (surface grain size 0.2; channel threshold 115, sphericity filter <0.85), which was followed by manual selection of all surfaces corresponding to cilia morphology. As a proxy of ciliary length, the longest principal axis of a minimal rectangular box, which fully encloses the object, was used ("BoundingBoxOO Length C" in Imaris). Categories of morphology of cilia were assigned manually, using three-dimensional images. Cilia were analyzed from six fields of view per mouse. The analyses were carried out blindly to genotypes.

## Transmission electron microscopy

Mice were euthanized using $CO_2$. 1–2 mm³ pieces of cortex were dissected out and fixed overnight at 4°C with 2.5% glutaraldehyde (EM-grade, Sigma-Aldrich check in the lab) in 0.1 M sodium phosphate buffer, pH 7.4. After washing, samples were post-fixed with 1% non-reduced osmium tetroxide in 0.1 M sodium phosphate buffer for 1 h at room temperature, dehydrated in ethanol series, and acetone prior to gradual infiltration into Epon (TAAB 812). After polymerization at 60°C for 18 h,

ultrathin sections were cut and post-stained with uranyl acetate and lead citrate. TEM micrographs were acquired with a Jeol JEM-1400 microscope (Jeol Ltd) running at 80 kV using an Orius SC 1000B camera (AMETEK Gatan, Inc.).

## Scanning electron microscopy

Mice were transcardially perfused with ice-cold PBS followed by perfusion with ice-cold 4% formaldehyde solution in PBS. Brains were postfixed in 2% formaldehyde solution in 0.1 M sodium phosphate buffer for 2–6 more hours at room temperature. En-face whole-mounts of the lateral ventricle wall were dissected (Mirzadeh et al., 2010), and postfixed in 2.5% glutaraldehyde in 0.1 M sodium phosphate buffer. Samples were stained in 2% non-reduced osmium tetroxide in 0.1 M sodium phosphate buffer, pH 7.4 for 2 h, dehydrated in ascending concentrations of ethanol, and dried by critical point drying (Leica EM CPD300, Leica Mikrosystems). The samples were oriented using dissection microscope, mounted on aluminum stubs covered with carbon tape, and coated with a thin layer of platinum. Scanning electron microscopy micrographs were taken under high vacuum using a FEI FEG-SEM Quanta 250 (Thermo Fisher Scientific) with a 5.00 kV beam, spot size 3.5.

## In vivo labeling of cilia

To label cilia in vivo in the mouse brain, pups (P0-P5) were injected intraventricularly with AAV-gfaABC1D-Arl13b-eGFP viral vector. pAAV-ABC1D-Arl13b-eGFP plasmid was subcloned using destination vector pAAV-GFAP-GFP (50473; Addgene), and a PCR-amplified insert with overhangs complimentary to destination vector from pLi3-Arl13b-eGFP plasmid (40879; Addgene [Larkins et al., 2011]). Assembly was done using NEB HiFI DNA assembly kit (NEB E5520) according to manufacturer's instructions. The plasmid was validated by direct sequencing of the insert fragment. For AAV production, the plasmid was amplified using chemically competent cells deficient for recombinase activity (#200152; Agilent Technologies), purified (#740416.10; Macherey-Nagel), and packaged into AAV8 vector (packaging was done by University of Helsinki AAV core unit). For injections, pups were cryoanesthetized and injections were done using stereotaxic frame; 2 µl per hemisphere was administered, the AAV titer was 3–3.5 × 10e−9 vp/µl.

## In situ enzyme activity of oxidative phosphorylation complexes

The histochemical activity assay was done as described previously (Forsström et al., 2019). Mice were euthanized with $CO_2$, freshly collected brains were embedded in embedding media (O.C.T. compound #4583; Tissue-Tek) and frozen in 2-methylbutane bath under liquid nitrogen cooling. For simultaneous colorimetric detection of the activity of Complex IV (cytochrome-c-oxidase [COX]) and Complex II (succinate dehydrogenase [SDH]), 12 µm brain sections were incubated with enzyme substrates (30 min for COX, room temperature), 40 min for SDH (+37°C). COX: 0.05 M phosphate buffer (pH 7.4) with 3,3–diaminobenzidine (DAB), catalase, cytochrome c and sucrose. SDH: 0.05 M phosphate buffer (pH 7.4) with nitro-blue tetrazolium and sodium succinate. Sections were then dehydrated by incubations in ascending concentrations of ethanol, xylene-treated and mounted.

## RT-qPCR

RNA from purified astrocytes or brain cell suspension after enzymatic digestion was extracted using RNA-binding columns (NucleoSpin RNA plus #740984.250; Macherey-Nagel), genomic DNA was eliminated, and cDNA was synthetized (#K1672; Thermo Fisher Scientific). Reverse transcription quantitative PCR (RT-qPCR) reactions were performed (#BIO-98020; Bio-line). NCBI primer BLAST software was used for oligonucleotide design (Ye et al., 2012), 70–150-bp-long product was amplified, and oligonucleotides were designed to be separated by at least one intron of the corresponding genomic DNA of a minimum 1,000 bp length (whenever possible). Oligonucleotides for amplification of Atp1b2, Slc1a3, Cspg4, Ocln are from Batiuk et al. (2017) and Syt1 from Liddelow et al. (2017). Oligonucleotide sequences can be found in Table S5. The expression of the genes of interest was normalized to hydroxymethylbilane synthase (Hmbs) or tyrosine 3-monooxygenase/tryptophan 5-monooxygenase activation protein zeta (Ywhaz) expression, the expression levels of which were not changed in TwKO[astro] cortical lysates or purified astrocytes compared to controls.

## mtDNA quantification

Purified astrocytes or brain tissue suspension after enzymatic digestion was lysed in TNES buffer with Proteinase K overnight at 55°C water bath. DNA was extracted using phenol–chloroform, followed by ethanol and ammonium acetate precipitation. Pellet was washed with ethanol, dried, and resuspended in 10 mM Tris-Cl, pH 8.0. Quantitative analysis of mtDNA was performed by quantitative PCR, normalized to a nuclear gene Rmb15 as described in Ignatenko et al. (2020). Oligonucleotide sequences can be found in Table S5.

## Metabolomics

Metabolomics datasets are from Ignatenko et al. (2020). All metabolites belonging to lipids and lipid-like molecules (HMDB super class) were analyzed and grouped for plotting according to HMDB classification. The mice were 2.3 mo old. See also Table S2.

## RNA sequencing

RNA was extracted from purified astrocyte preparations using RNA-binding columns (NucleoSpin RNA plus #740984.250; Macherey-Nagel). RNA was analyzed using TapeStation, and only samples with RIN > 7 were used. RNA sequencing was performed in the BGI Genomics, China. mRNA was enriched by Oligo dT selection, and reverse transcription was carried out using random N6 primer. End repair and adaptor ligation were followed by PCR amplification of the library. The library was sequenced using BGISEQ-500 platform, paired-end sequencing, and read length 100 bp. After sequencing, reads were filtered to remove adaptor sequences, contamination, and low-quality reads using Soapnuke software version 1.6.7 (filtering parameters -n 0.001 -l 20 -q 0.4 -A 0.25). Additionally, quality of reads was analyzed using FastaQC software. The data are available in

NCBI GEO repository (accession no. GSE174343; Edgar et al., 2002). See also Table S6.

### Proteomics (LC-ESI-MS/MS analysis)

Mass spectrometry analyses were performed at the Turku Proteomics Facility, Finland, similarly to Callister et al. (2006); Huang et al. (2020). The proteins of enriched astrocyte cell fraction were subjected to in-solution digestion. After digestion, peptides were desalted with a Sep-Pak C18 96-well plate (Waters), evaporated to dryness, and stored at −20°C. Samples were analyzed by a data-independent acquisition (DIA) LC-MS/MS method using a QExactive HF mass spectrometer. Data were analyzed by Spectronaut software. Digested peptide samples were dissolved in 0.1% formic acid and peptide concentration was determined with a NanoDrop device. For DIA analysis, 800 ng peptides was injected and analyzed in a random order. Wash runs were submitted between each sample to reduce potential carry-over of peptides. The LC-ESI-MS/MS analysis was performed on a nanoflow HPLC system (Easy-nLC1000, Thermo Fisher Scientific) coupled to the QExactive HF mass spectrometer (Thermo Fisher Scientific) equipped with a nano-electrospray ionization source. Peptides were first loaded on a trapping column and subsequently separated inline on a 5 cm C18 column (75 μm × 15 cm, ReproSil-Pur 3 μm 120 Å C18-AQ, Dr. Maisch HPLC GmbH, Ammerbuch-Entringen). The mobile phase consisted of water with 0.1% formic acid (solvent A) or acetonitrile/water (80:20 [v/v]) with 0.1% formic acid (solvent B). A 110 min gradient was used to elute peptides (70 min from 5 to 21% solvent B followed by 40 min from 21% to 36 min solvent B). Samples were analyzed by a data independent acquisition (DIA) LC-MS/MS method. MS data were acquired automatically using Thermo Xcalibur 4.1 software (Thermo Fisher Scientific). In a DIA method, a duty cycle contained one full scan (400–1,000 m/z) and 40 DIA MS/MS scans covering the mass range 400–1,000 with variable width isolation windows.

### Bioinformatic analyses
#### RNA sequencing

Quantification of transcript abundance in RNA sequencing dataset was performed using Kallisto software (mm10 build [Bray et al., 2016]). Differential expression analysis and PCA plot were prepared with Sleuth (Pimentel et al., 2017), gene ontology pathway enrichment analysis was carried out with Metascape (Zhou et al., 2019).

#### Upregulated gene list

Top 1,000 genes were analyzed (genes with q-value < 0.1 and log2(FC) > 0.3 were selected, and then sorted by log2(FC)).

#### Downregulated gene list

All genes with q-value <0.1 and log2(FC) < −0.3 were selected (408 genes). RNA sequencing analysis of published datasets (Table S3): available from NCBI GEO repository (GSE76097; GSE99791; GSE96518; GSE143598; GSE125610): log2(FC) from the datasets were used (Anderson et al., 2016; Boisvert et al., 2018; Guttenplan et al., 2020; Kühl et al., 2017), or calculated from raw read counts using DESeq2 (Li et al., 2019; Love et al., 2014).

For motif enrichment analysis (Table S4), genomic sequences corresponding to 500 bp upstream and 500 bp downstream from the transcription start sites (TSS) of genes from upregulated gene list were extracted (log2(FC) > 0.3 and P-adj-value <0.05; mm10 build). As a negative dataset, 1,000 randomly selected genes from the mouse genome were used. To find enriched motifs, CentriMo tool from the MEME Suite was used (Bailey and Machanick, 2012). For analysis of TF-binding site enrichment, genomic coordinates for binding sites of RFX1-3 based on a previously published ChIP-Seq study were used (Lemeille et al., 2020). Genomic coordinates spanning 10,000 bp upstream and 10,000 bp downstream from TSSs of upregulated genes (log2(FC) > 0.3 and P-adj-value <0.01) were extracted. An overlay of the two datasets revealed the RFX-binding sites within TSS proximal sequences of upregulated genes. As negative datasets either randomly selected genes from the mouse genome or genes which were expressed in astrocytes but not changed based on our RNA-Seq results were used. GRCm38 (mm10) mouse genome build was used for all analyses and manipulations on genomic intervals were carried out using the Galaxy platform (Afgan et al., 2018).

#### Proteomics

Swiss-Prot 2021_04 Mus Musculus was used as a reference database, with precursor FDR Cutoff = 0.01 and protein FDR Cutoff = 0.01. Quantification type: area under the curve within integration boundaries for each targeted ion. Normalization: local normalization (based on RT dependent local regression model; Callister et al., 2006). Differential abundance analysis: Unpaired Student's *t* test with combined MS1 + MS2 statistical model (Huang et al., 2020) and multiple-testing correction of the P values (q-values).

#### Statistical analysis

Statistical analyses were performed using Prism 8 software and R; graphs were made with Prism 8. Statistical analysis of differential gene expression of RNA sequencing data generated in this study was performed using the Sleuth package (Pimentel et al., 2017), q-values were used throughout the study to determine statistical significance of these comparisons. Statistical analyses of the cilia length distribution were carried out using Kolmogorov–Smirnov test in Prism 8 and ANOVA test in R base software. Statistical analysis of cilia morphology and RNA-fluorescence in situ hybridization between two groups was performed using unpaired two-tailed parametric *t* test in Prism 8. Heatmaps were generated using Complex Heatmap package in R (Gu et al., 2016), Z-scores were calculated from scaled reads per base. Data distribution was assumed to be normal but this was not formally tested.

#### Online supplemental material

Fig S1 is related to Figs 1 and 3 and shows histological consequences of Twinkle KO in mouse astrocytes. Fig. S2 is related to Figs 2 and 4 and shows transcriptomic consequences of Twinkle KO in purified cortical astrocytes. Table S1 shows reactive astrogliosis markers. Table S2 shows lipids and lipid-like molecules. Table S3 lists cilia genes. Table S4 shows ChIP Seq

analysis. Table S5 shows oligonucleotides used in the study. Table S6 shows RNA sequencing and proteomics TwKO^astro vs. Control purified adult astrocytes.

## Acknowledgments

We thank Brendan Battersby and Maxim Bespalov for insightful discussions; Markus Innilä, Kirsi Mattinen, Mervi Lindman, Tuula Manninen, Babette Hollmann, and Sonja Jansson for technical assistance; Cory Dunn for sharing materials; Ilya Belevich and Antti Isomäki for technical expertise. We acknowledge the core facilities of the University of Helsinki (UH): Biomedicum Imaging Unit, Genome Biology Unit, AAV core, and Laboratory Animal Center; and Turku Proteomics Facility.

We thank the following funding sources: Academy of Finland, Sigrid Jusèlius Foundation, Jane and Aatos Erkko Foundation, UH (A. Suomalainen); UH Doctoral Program in Biomedicine, and Biomedicum Foundation, Otto Malm Foundation, Oskar Öflunds Foundation, Maud Kuistila Foundation (O. Ignatenko); European Molecular Biology Organization (ALTF 1185-2017), Human Frontier Science Program Organization (LT000446/2018-L) (J. Nikkanen); UH Doctoral Program Brain and Mind (S. Rybas).

The authors declare no competing financial interests.

Author contributions: O. Ignatenko, G. Ince-Dunn, and A. Suomalainen conceived and conceptualized the research. O. Ignatenko designed, planned, and executed all main experiments, analyzed and interpreted data. S. Malinen and S. Rybas executed experiments, analyzed, and interpreted data. O. Ignatenko, A. Kononov, J. Nikkanen, G. Ince-Dunn executed and interpreted bioinformatic analyses. A. Kononov provided expertise for statistical analyses. H. Vihinen, E. Jokitalo supervised electron microscopy experiments. O. Ignatenko assembled figures and wrote the first draft of the manuscript. O. Ignatenko, G. Ince-Dunn, and A. Suomalainen wrote the manuscript, and all authors commented. G. Ince-Dunn and A. Suomalainen supervised the study.

Submitted: 9 March 2022

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

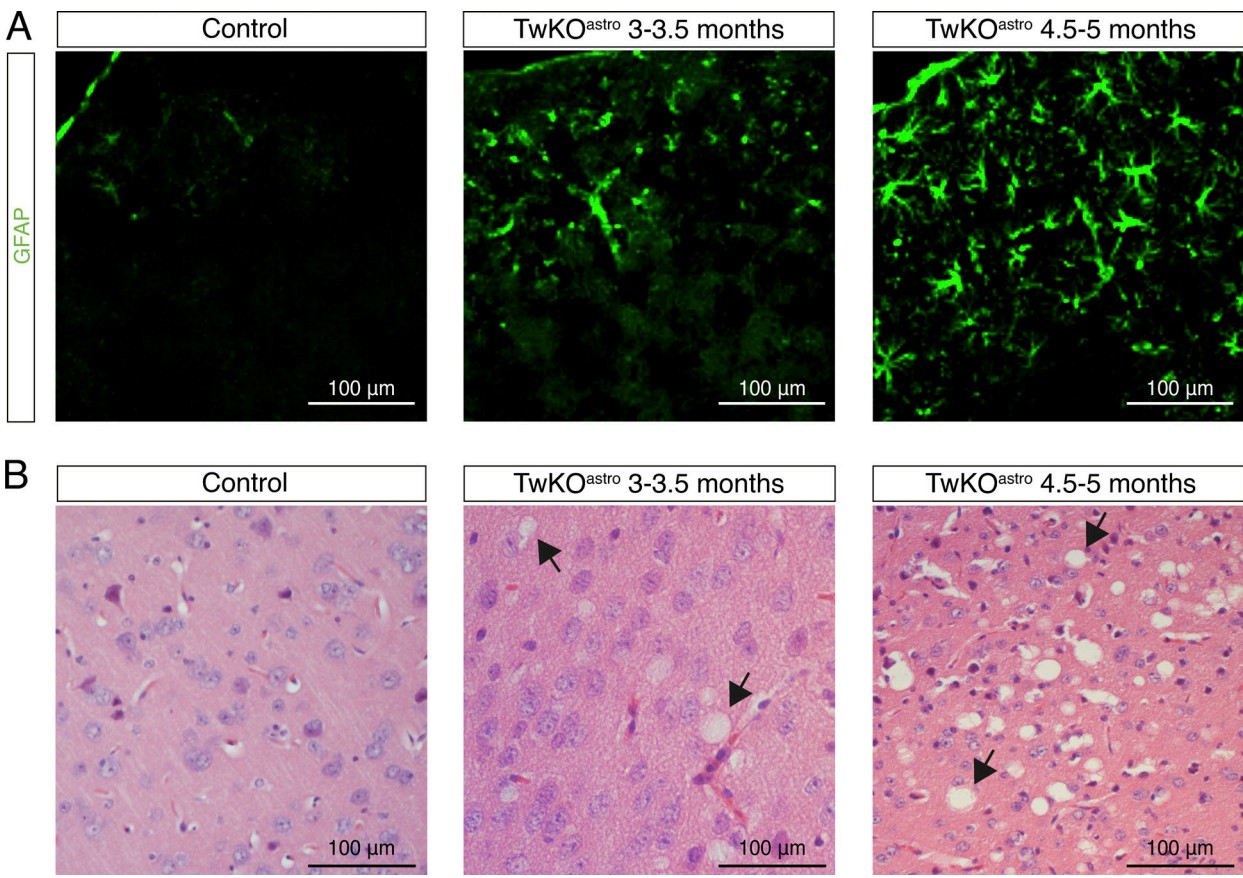

Figure S1.   **Related to** Figs. 1 and 3**. Histological consequences of Twinkle KO in mouse astrocytes. (A)** Reactive gliosis in cerebral cortex of TwKO[astro] mice. Immunostaining against GFAP, images are representative of five mice per genotype. **(B)** Spongiotic encephalopathy in cerebral cortex TwKO[astro] mice, images are representative of five mice per genotype. Hematoxylin and eosin staining. Arrows indicate examples of spongiotic pathology.

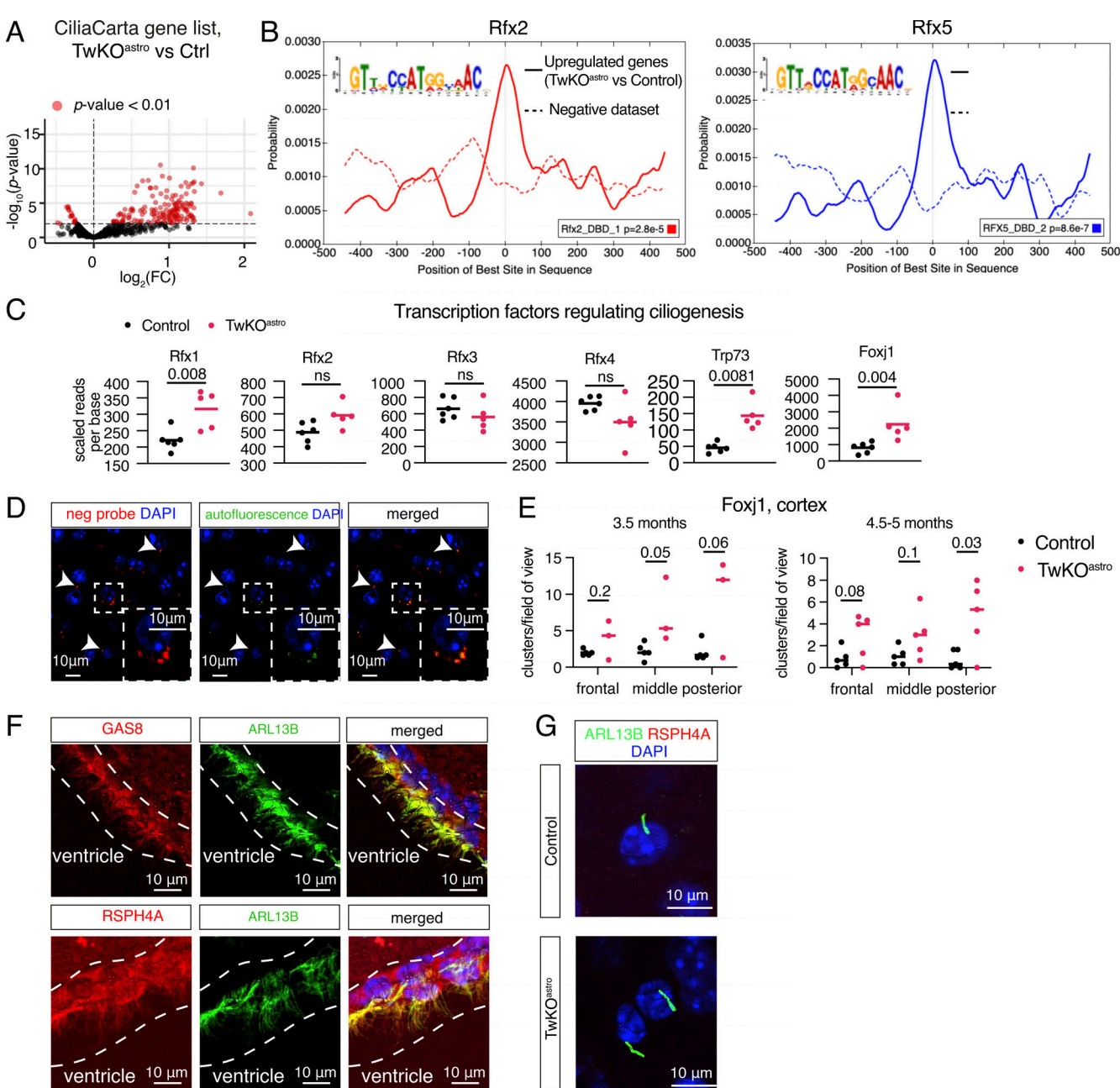

Figure S2.   **Related to** Figs. 2 and 4. **Transcriptomic consequences of Twinkle KO in purified cortical astrocytes. (A)** Expression levels of ciliary genes (CiliaCarta gene list [van Dam et al., 2019]). See also Fig. 2 A. **(B)** Motif enrichment analysis of promoters of upregulated genes. Motif probability graph. See also Fig. 2 C. **(C)** Expression of transcription factors that regulate ciliogenesis, scaled reads per base. Expression data for these genes are also presented in Fig. 2, E and F. P values calculated using unpaired two-tailed parametric t test. **(D)** Control experiment for Foxj1, RNA-fluorescence in situ hybridization with the negative probe. Arrowheads: Puncta in the red channel (negative probe) that overlap with the green channel (autofluorescence) were excluded from analysis. Inset: An enlarged image. (Unrelated to this analysis, these mice also express AAV-gfaABC1D-Arl13b-eGFP.) See also Fig. 2 H. **(E)** Quantification of Foxj1 RNA-fluorescence in situ hybridization in the mouse brain cortex. Signal quantified from three fields of view per mouse, symbols represent the average per mouse. Clusters with a minimum of three puncta were quantified. Control n = 5, TwKOastro n = 3. See also Fig. 2 H. **(F)** Motile cilia components GAS8 and RSPH4A in ependymal multiciliated cells of control mouse brain, co-stained with ARL13B; immunofluorescence. Dashed lines indicate the ependymal cell layer. **(G)** Motile cilia component RSPH4A immunostained with ARL13B, brain cortex of 4.5–5-mo-old mice. **(A–C)** RNA sequencing, astrocytes purified from TwKOastro compared to control mice. The dataset is described in Fig. 1. FC, fold change (A). See also Table S3.

**Provided online are Table S1, Table S2, Table S3, Table S4, Table S5, and Table S6. Table S1 shows reactive astrogliosis markers. Table S2 shows lipids and lipid-like molecules. Table S3 lists cilia genes. Table S4 shows ChIP Seq analysis. Table S5 shows oligonucleotides used in the study. Table S6 shows RNA sequencing and proteomics TwKOastro vs. Control purified adult astrocytes.**

