## [Peer Review File · The Journal of Cell Biology]

Mitochondrial dysfunction compromises ciliary homeostasis in astrocytes

Olesia Ignatenko, Satu Malinen, Sofii Rybas, Helena Vihinen, Joni Nikkanen, Aleksandr Kononov, Eija Jokitalo, Gulayse Ince Dunn, and Anu Suomalainen-Wartiovaara

Corresponding Author(s): Anu Suomalainen-Wartiovaara, University of Helsinki and Gulayse Ince Dunn, University of Helsinki

Review Timeline:

Submission Date:	2022-03-09
Editorial Decision:	2022-04-20
Revision Received:	2022-08-19
Editorial Decision:	2022-09-26
Revision Received:	2022-10-06

Monitoring Editor: Richard Youle

Scientific Editor: Dan Simon

Transaction Report:

DOI: <https://doi.org/10.1083/jcb.202203019>

Revision 0

Review #1

1. Evidence, reproducibility and clarity:

Evidence, reproducibility and clarity (Required)

This well-written and convincing manuscript surprisingly links for the first time mitochondrial dysfunction to cilia homeostasis in a specific cell type of the brain, the astrocyte. Previously the authors have beautifully demonstrated that deletion of the Twinkle helicase in astrocytes leads to a more severe phenotype than the deletion of the gene in neurons, resembling human spongiform encephalopathy. Furthermore, they showed that mitochondrial dysfunction elicited in astrocytes an integrated stress response characterized by specific features. Now they go a step further, by purifying astrocytes deleted of Twinkle and analyzing their global transcriptional profile. Surprisingly, they detect an upregulation of genes linked to ciliary function, especially to the motile cilium. This response is caused by the activation of the FOXJ1 and RFX transcription factors, which are known master regulators of motile ciliogenesis. Strikingly, cilia of Twinkle-deficient astrocytes appeared abnormal, being elongated or with contorted morphology. This phenotype was not observed in ependymal cells that are also targeted by the Cre line used by the authors, indicating cell specificity. The authors propose that this response is an astrocyte-specific anabolic response due to the ISR activation.

Overall, this is an original study that opens a new paradigm: the possible implication of abnormal ciliogenesis as a cell-specific response to mitochondrial dysfunction. The limitation of this study, as the authors openly discuss, is that it is unclear what is the pathological significance of this abnormal ciliary program. Unravelling the functional significance of this pathway will however require additional years of study.

****Minor comments:****

1. The quantification of the beautiful ciliary phenotype is a crucial result of this study. More explanation should be provided in the material and methods about the method used to quantify the length of the cilium. It seems difficult to quantify cilia with contorted morphology. The authors should also specify if the analyses were conducted blind to the genotype.
2. The lipidomic data are certainly interesting, but it is not clear why the authors decide to selectively show these data (from the dataset described in the previous Ignanenko et al., 2020 paper). The link to the previous part of the manuscript and the relevance for the abnormal cilia formation is not clear. Also the statement that the abnormal lipid profiles depend on alteration in beta-oxidation is just a speculation. This can be more clear in the discussion.
3. Figure S2G: The authors could discuss why cultured astrocytes deprived of HBEGF or treated with EGFRi show a similar response as observed here. Is it possible to find some parallelism, for example in the metabolic response?
4. Figure 2G does not allow to compare controls and Twinkle KO. Can the authors compare the

data in a single graph?

5. For promoter analysis, have the authors used all upregulated genes or only those linked to ciliary programs? Is the RFX motif found also in other ISR-driven genes?

2. Significance:

Significance (Required)

An important aspect explaining the specificity of the disease phenotype in case of mitochondrial dysfunction is the tissue-specificity of elicited stress response pathways. This manuscript provides an entirely novel point of view, implicating abnormal ciliogenesis as a novel target of this response specifically in astrocytes, which are important players in the pathogenesis of mitochondrial brain disease. In the context of several papers now showing stress responses upon mitochondrial dysfunction, this paper highlights a novel and totally unexpected pathway. I expect that the audience for this manuscript will be far beyond scientists working in the field of mitochondrial dysfunction and neurodegeneration, like this reviewer. I expect that these findings will also appeal to the currently rather separate field of cilia function and brain disease.

3. How much time do you estimate the authors will need to complete the suggested revisions:

Estimated time to Complete Revisions (Required)

(Decision Recommendation)

Less than 1 month

Review #2

1. Evidence, reproducibility and clarity:

Evidence, reproducibility and clarity (Required)

In this manuscript by Ignatenko et al., the authors follow up on two previous works from the same authors, using mice with a defect in mitochondrial function specifically in astrocytes. While works in recent years interrogated the effects of knocking out mitochondrial genes mainly in peripheral tissues (e.g. metabolic organs), the current work focuses on astrocytes. These non-neuronal cells of the nervous system have been less studied in the context of mitochondrial defects. In the current work, the authors focus on exploring the transcriptional response in astrocytes to the mitochondrial insult, which revealed induction of cilia-related processes, similar to the findings of Bae et al 2019 (10.1038/s41419-019-2184-y) in cultured cells. This work provides conceptual insight and advancement at the level of cell biology in an in-vivo model,

illustrating a morphological and transcriptional change in astrocytes upon mitochondrial stress. The work described by Ignatenko et al. will be interesting to the field of mitochondrial biology, and researchers in cell biology, as it shows that the ciliogenic changes occur also in an in-vivo model.

The authors' work is comprehensive, rigorous, and elegant. Experiments are well-controlled and the statements in the manuscript are generally convincing considering the data presented in the figures. The manuscript is beautifully written and easy to follow. While the researchers did a terrific job, there are two points that could be improved. First - the main point, that mitochondrial dysfunction triggers changes in cilium-related processes, is mostly supported by gene expression changes, and the validation experiment relies solely on the pan-cilia marker ARL13B. The authors show that a transcriptional program of motile cilia components is induced, however this was not experimentally verified beyond RNA levels (see below). Finally, the only change observed in cilia upon the mitochondrial defect is morphological, with no clear functional outcomes.

Data supporting claims on motile cilia

Previous works showed that astrocytes have one single primary cilia. In the current work, the authors observe induction of motile cilia genes. However, they do not address directly, experimentally, whether these results signify a switch between primary and motile mitochondria, or whether this is simply an induction of a dysfunctional ciliogenic program, that thus creates the morphological defects they observe (bent/contorted cilia). The authors follow gene-expression changes, but it is not clear whether the components they see induced in fact get incorporated into the cilia (this could be examined, for example, for genes identified in their dataset using antibodies).

In addition, in the discussion they hypothesize three different mechanisms, however this gap is important to provide context for the significance of the results. If indeed there is a switch towards more motile-related functions, this needs to be examined. If it is not possible to measure cilia function in this system, the authors should expand their discussion to clearly address the possibly scenarios, and provide functional context, considering the functional effects seen in similar cell culture experiments (Bae et al.), or in other systems where bent/contorted cilia were observed.

Finally, the authors rely only on staining ARL13B to identify cilia in astrocytes, however they use additional methods (Figure 4, scanning electron microscopy) for ependymal cells when examining cilia in other cells. Why was SEM not used to verify their results for astrocytes, the main cell type discussed in the manuscript? If this is experimentally possible, this should be done, as it does not rely on protein markers and antibody, which also show some staining of other cellular components in the manuscript (Figure 3).

Tissue specificity and stress specificity in triggering cilia

Figure S2G is an important figure addressing the generalizability of these results. Importantly, Bae et al. showed that other mitochondrial stresses can also trigger the ciliogenic response. To improve the impact of the current work, the authors should deepen and expand this section, using any other gene expression datasets that are relevant and available (other stresses, knockout of Twinkle in other tissues, different organisms, etc). In addition, the authors can easily define a

metric to represent the data to better give a sense whether a positive or negative (more or less cilia-related processes) effect is observed, in addition to the heatmap, that is difficult to interpret but can be provided as a supplementary.

****Minor comments:****

1. The last section of the results, discussing metabolic and lipid changes in response to the mitochondrial stress, seems out of place. At the current version, there is no conceptual connection between the cilia data (Figure 1-4) and the metabolic section (supplementary figure 4), for which, data is largely published in a previous work, and re-analyzed. To possibly improve this, the authors may consider either (1) moving Figure S4 to Figure 1 or S1, and re-write the first section of the results. This may be presented as surveying the effects of the mitochondrial dysfunction using both their previously published metabolomic data, along with their new astrocyte-specific RNA-seq. (2) Figure S4 and the accompanying text may be eliminated from the manuscript.
2. Some of the methods sections should be expanded, and methods should be explained in more length rather than citing previous works (e.g. astrocyte sorting, FISH) including experimental details (concentrations, dilutions, incubations, etc) to allow reproducibility.
3. Number of biological repeats needs to be mentioned for each method/experiment.
4. Why wasn't GFAP was used as one of the markers for 1A?
5. There is some enrichment (~2 fold) in endothelial markers, the authors should explain in the text this caveat.
6. The authors state "These results provided a proof of principle that the purified fraction represented the astrocyte population with Twnk-KO". Please revise the sentence, as these results "suggest/show/demonstrate" that the purified fraction represents the astrocyte population. This is especially true due to the inter-cellular effects that are known to occur upon mitochondrial stress, which may affect the levels of these genes in other tissues, despite not having the Twnk-KO within the same cells.
7. Figure 3E reference in the text has a mistake - "Contorted cilia in TwKOastro included S-shaped and corkscrew-like morphologies, and occasionally the long cilia appeared to form several loops (Fig. 3F, bottom panel)". (there is no bottom panel).
8. The text relating to Figure S2F "Finally, among factors contributing to primary cilia signaling, the expression of the sonic hedgehog pathway effector Gli1 was induced in astrocytes sorted from TwKOastro mice (Fig. S2F)" seems misplaced in terms of the logic of the paper. Consider moving to an earlier section, or explaining why this is relevant in its current mention location.
9. Authors are missing a reference for another study that examined astrocytes and mitochondrial stress, 10.1016/j.cmet.2020.05.001, which they should cover and cite.

2. Significance:

Significance (Required)

See comments above.

3. How much time do you estimate the authors will need to complete the suggested revisions:

Estimated time to Complete Revisions (Required)

(Decision Recommendation)

Between 1 and 3 months

Review #3

1. Evidence, reproducibility and clarity:

Evidence, reproducibility and clarity (Required)

In this paper, Ignatenko, et al. describe a set of experiments demonstrating the link between mitochondrial dysfunction in astrocytes and motile ciliogenesis. Using astrocytes isolated from TwKOastro mice, which possess a knockout of the mitochondrial DNA helicase Twinkle in astrocytes leading to reactive gliosis, the researchers provide results from several experiments to support their central claim that respiratory chain deficiency in astrocytes activates FOXJ1 and RFX transcription factors which results in changes to motile ciliogenesis. Using RNASeq, the researchers demonstrated mtDNA mutations impact motile cilia in astrocytes taken from TwKOastro mice. Previously reported ChIPSeq data from ependymal cells was then used to determine which transcription factor sequences were enriched in TwKOastro mice. Foxj1 expression was observed in astrocytes, leading researchers to hypothesize that mitochondrial damage in astrocytes caused by the deletion of Twinkle leads to an increase in Foxj1 expression, producing an increase in motile ciliogenic program in astrocytes. Scanning electron microscopy demonstrated that the primary cilium of TwKOastro mice is deformed compared to control. Additionally, the researchers demonstrated that this response to mitochondrial dysfunction is cell-type specific to astrocytes by comparing the results from the above experiments to other models of mitochondrial dysfunction in other cell types. These findings contribute more information to the field of astrocyte involvement in the pathogenesis of neurodegenerative disorders. Additionally, these experiments demonstrate previously unknown roles of RFX and FOXJ1 in adult astrocytes and in integrated stress response in the mitochondria. Paper is well done and properly interpreted! It is also one of the first to mechanistically link mitochondrial function to astrocytes.

****Major Comments:****

- One thing that would be important if possible is metabolomics. For example, is aspartate/asparagine levels altered that could trigger ISR. Also, whether L-2HG, fumarate or succinate levels ratio to aKG differ, as this ratio can modulate aKG dioxygenases like TETs.

Metabolomics with the RNA seq would be helpful. But I realize it could be challenging from in vivo astrocytes.

****Minor Comments****

- In Figure 3F, the authors should change the pie chart into an easier to read data visualization, like a stacked bar graph. As is, it is difficult to clearly see the differences between groups on a pie chart.
- The authors should include an experiment looking at *Foxj1* expression levels via RT-qPCR in wild-type astrocytes, not only GFAP-Cre mice.

2. Significance:

Significance (Required)

There is very little known about how mitochondrial function controls astrocytes. paper is one of the first!

3. How much time do you estimate the authors will need to complete the suggested revisions:

Estimated time to Complete Revisions (Required)

(Decision Recommendation)

Between 1 and 3 months

Revision Plan

Manuscript number: RC-2022-01235

Corresponding author(s): Gulayse, Ince Dunn; Anu, Suomalainen

1. General Statements [optional]

Au: We thank for the highly favourable and enthusiastic comments of the three reviewers, and their highly supportive assessment of our work. All of them appreciated the conceptual novelty of our study. They suggested minor improvements, which we have addressed point-by-point below.

We have previously shown that mitochondrial spongiotic encephalopathy, a common and devastating manifestation of patients with mitochondrial brain disease, can be caused by pathologically reactivated astrocytes. Here, we report novel molecular mechanisms of astrocyte reactivation as a consequence of mitochondrial dysfunction, specifically, mtDNA depletion. Our transcriptomics analyses from purified cortical astrocytes led to the identification of an unexpected and novel pathway: a wide activation of motile ciliogenesis program in astrocytes that are known to harbor single primary cilium. We further carried out detailed functional and morphological experiments in vivo to reveal that an aberrant upregulation of a motile ciliogenesis program was associated with primary cilia elongation and abnormal morphology, indicating that mitochondrial brain diseases have a component of metabolic ciliopathy.

Reviewer #1 (Evidence, reproducibility and clarity (Required)):

This well-written and convincing manuscript surprisingly links for the first time mitochondrial dysfunction to cilia homeostasis in a specific cell type of the brain, the astrocyte. Previously the authors have beautifully demonstrated that deletion of the Twinkle helicase in astrocytes leads to a more severe phenotype than the deletion of the gene in neurons, resembling human spongiotic encephalopathy. Furthermore, they showed that mitochondrial dysfunction elicited in astrocytes an integrated stress response characterized by specific features. Now they go a step further, by purifying astrocytes deleted of Twinkle and analyzing their global transcriptional profile. Surprisingly, they detect an upregulation of genes linked to ciliary function, especially to the motile cilium. This response is caused by the activation of the FOXJ1 and RFX transcription factors, which are known master regulators of motile ciliogenesis. Strikingly, cilia of Twinkle-deficient astrocytes appeared abnormal, being elongated or with contorted morphology. This phenotype was not observed in ependymal cells that are also targeted by the Cre line used by the authors, indicating cell specificity. The authors propose that this response is an astrocyte-specific anabolic response due to the ISR activation.

Overall, this is an original study that opens a new paradigm: the possible implication of abnormal ciliogenesis as a cell-specific response to mitochondrial dysfunction. The limitation of this study, as the authors openly discuss, is that it is unclear what is the pathological significance of this abnormal ciliary program. Unravelling the functional significance of this pathway will however require additional years of study.

Au: We thank the reviewer for the positive evaluation of our work and thoughtful summary of findings.

Revision Plan

Reviewer #1 (Significance (Required)):

An important aspect explaining the specificity of the disease phenotype in case of mitochondrial dysfunction is the tissue-specificity of elicited stress response pathways. This manuscript provides an entirely novel point of view, implicating abnormal ciliogenesis as a novel target of this response specifically in astrocytes, which are important players in the pathogenesis of mitochondrial brain disease. In the context of several papers now showing stress responses upon mitochondrial dysfunction, this paper highlights a novel and totally unexpected pathway. I expect that the audience for this manuscript will be far beyond scientists working in the field of mitochondrial dysfunction and neurodegeneration, like this reviewer. I expect that these findings will also appeal to the currently rather separate field of cilia function and brain disease.

Au: We thank the reviewer for enthusiastic reading of our work and seeing potential in its impact.

Reviewer #2 (Evidence, reproducibility and clarity (Required)):

In this manuscript by Ignatenko et al., the authors follow up on two previous works from the same authors, using mice with a defect in mitochondrial function specifically in astrocytes. While works in recent years interrogated the effects of knocking out mitochondrial genes mainly in peripheral tissues (e.g. metabolic organs), the current work focuses on astrocytes. These non-neuronal cells of the nervous system have been less studied in the context of mitochondrial defects. In the current work, the authors focus on exploring the transcriptional response in astrocytes to the mitochondrial insult, which revealed induction of cilia-related processes, similar to the findings of Bae et al 2019 (10.1038/s41419-019-2184-y) in cultured cells. This work provides conceptual insight and advancement at the level of cell biology in an in-vivo model, illustrating a morphological and transcriptional change in astrocytes upon mitochondrial stress. The work described by Ignatenko et al. will be interesting to the field of mitochondrial biology, and researchers in cell biology, as it shows that the ciliogenic changes occur also in an in-vivo model.

The authors' work is comprehensive, rigorous, and elegant. Experiments are well-controlled and the statements in the manuscript are generally convincing considering the data presented in the figures. The manuscript is beautifully written and easy to follow.

Au: We thank the reviewer for such favorable assessment our work.

Reviewer #2 (Significance (Required)):

See comments above.

Reviewer #3 (Evidence, reproducibility and clarity (Required)):

In this paper, Ignatenko, et al. describe a set of experiments demonstrating the link between mitochondrial dysfunction in astrocytes and motile ciliogenesis. Using astrocytes isolated from TwKOastro mice, which possess a knockout of the mitochondrial DNA helicase Twinkle in astrocytes leading to reactive gliosis, the researchers provide results from several experiments to support their central claim that respiratory chain deficiency in astrocytes activates FOXJ1 and RFX transcription factors which results in changes to motile ciliogenesis. Using RNASeq, the researchers demonstrated mtDNA mutations impact motile cilia in astrocytes taken from TwKOastro mice. Previously reported ChIPSeq

Revision Plan

data from ependymal cells was then used to determine which transcription factor sequences were enriched in TwKOastro mice. Foxj1 expression was observed in astrocytes, leading researchers to hypothesize that mitochondrial damage in astrocytes caused by the deletion of Twinkle leads to an increase in Foxj1 expression, producing an increase in motile ciliogenic program in astrocytes. Scanning electron microscopy demonstrated that the primary cilium of TwKOastro mice is deformed compared to control. Additionally, the researchers demonstrated that this response to mitochondrial dysfunction is cell-type specific to astrocytes by comparing the results from the above experiments to other models of mitochondrial dysfunction in other cell types. These findings contribute more information to the field of astrocyte involvement in the pathogenesis of neurodegenerative disorders. Additionally, these experiments demonstrate previously unknown roles of RFX and FOXJ1 in adult astrocytes and in integrated stress response in the mitochondria. Paper is well done and properly interpreted! It is also one of the first to mechanistically link mitochondrial function to astrocytes.

Au: We thank the reviewer for the endorsement of our work.

Reviewer #3 (Significance (Required)):

There is very little known about how mitochondrial function controls astrocytes. paper is one of the first!

Au: We share the excitement expressed by the reviewer! Thank you!

2. Description of the planned revisions

Points raised by Reviewer #2:

1) While the researchers did a terrific job, there are two points that could be improved. First - the main point, that mitochondrial dysfunction triggers changes in cilium-related processes, is mostly supported by gene expression changes, and the validation experiment relies solely on the pan-cilia marker ARL13B. The authors show that a transcriptional program of motile cilia components is induced, however this was not experimentally verified beyond RNA levels (see below).

Finally, the only change observed in cilia upon the mitochondrial defect is morphological, with no clear functional outcomes.

Au: Previous published evidence for astrocyte cilia analysis are based entirely on ARL13B as a marker (Sterpka and Chen 2018; Kasahara et al. 2014; Sipos, Komoly, and Ács 2018; Khan et al. 2021). Another commonly used marker to visualize axonemal morphology is AC3, which in the brain displays higher signal intensity in neurons compared to astrocytes, and thus is not used for visualization of cilia in astrocytes ((Sipos, Komoly, and Ács 2018; Sterpka and Chen 2018); our unpublished observations).

We agree with the reviewer that our analysis of motile ciliogenesis program is based on RNA level changes. To investigate protein level changes in TwKO^{astro} brain, we previously attempted immunostainings for selected proteins. We have now ordered a panel of antibodies against motile cilia

Revision Plan

components to test these further. Single cilia are very small structures that are difficult to image from specific cell types in the 3D space of the brain, but we certainly hope that this experiment will shed information about protein level changes in the astrocyte cilia of our mouse model.

Finally, the functional consequences of ciliary pathology in astrocytes and changes to ciliary signaling pathways form a whole new area of research that we are very much interested in pursuing. However, this is a completely new line of experimentation involving generating new mouse lines, pharmacological interventions, and not in the scope of this article.

2) The authors follow gene-expression changes, but it is not clear whether the components they see induced in fact get incorporated into the cilia (this could be examined, for example, for genes identified in their dataset using antibodies).

Au: To address this, we will carry out a series of immunostainings on the brains of control and TwKO^{astro} mice using commercially available antibodies against motile cilia factors, which are induced in our dataset (please see more detailed explanation above).

3. Description of the revisions that have already been incorporated in the transferred manuscript

Points raised by Reviewer #1:

1) The quantification of the beautiful ciliary phenotype is a crucial result of this study. More explanation should be provided in the material and methods about the method used to quantify the length of the cilium. It seems difficult to quantify cilia with contorted morphology. The authors should also specify if the analyses were conducted blind to the genotype.

Au: We added details of the cilia length analysis to methods section. As a proxy of ciliary length, we used the longest principal axis of a minimal rectangular box, which encloses a cilium on confocal microscopy images. We agree with the reviewer that it is difficult to estimate the exact length of cilia with contorted morphology. For example, if a cilium is contorted and makes loops, the length of such cilia would be underestimated. The majority of cilia in our datasets are however straight or bent (Figure 2F), and thus the method of choice is adequate for the purpose to investigate the distribution of lengths. Cilia analysis was done blinded to the genotype, which we now indicated in the Methods section.

2) The lipidomic data are certainly interesting, but it is not clear why the authors decide to selectively show these data (from the dataset described in the previous Ignanenko et al., 2020 paper). The link to the previous part of the manuscript and the relevance for the abnormal cilia formation is not clear.

Revision Plan

Also the statement that the abnormal lipid profiles depend on alteration in beta-oxidation is just a speculation. This can be more clear in the discussion.

Au: We agree with the reviewer that changes to lipid metabolism and cilia maintenance at first sight do not seem to be connected. However, we believe that presenting both up-regulated and down-regulated genes in this manuscript provides a more holistic view to the readers. We chose to present the lipidomics data from our previous study so that the readers are presented side-by-side the astrocyte-specific gene expression changes and biochemical changes relating to lipid metabolism from TwKO^{astro} mice (unpublished data). The analysis of lipid metabolite levels is based on a dataset we previously published.

In the updated version of our manuscript, we have provided a general introduction to our RNAseq results, including the lipid metabolism findings, in Figure 1, and have focused exclusively on the cilia findings for the remainder of the manuscript.

We also modified our statement about beta oxidation to highlight that it is our hypothesis.

3) Figure S2G: The authors could discuss why cultured astrocytes deprived of HBEGF or treated with EGFRi show a similar response as observed here. Is it possible to find some parallelism, for example in the metabolic response?

Au: We agree that this is an interesting observation and we have now added a sentence highlighting the potential existence of overlapping molecular changes in mtDNA-depleted and EGF signaling-inhibited astrocytes (page 9, paragraph 1). Currently, we do not know whether there are overlapping mechanisms and what these might be in these two different means of perturbing astrocyte physiology. We would like to mention that some other studies of e.g. mouse models of familial Parkinson's disease and traumatic brain injury, have reported changes to cilia morphology (Khan et al. 2021; Sterpka et al. 2020; Moser, Fritzler, and Rattner 2009). The lack of transcriptomics data in these studies excluded them from our comparative analysis in Fig S2G. It will certainly be fascinating to uncover how and why astrocytes respond to specific insults (for example, by restructuring ciliary morphology or having impaired turnover of ciliary factors), and how this impacts the main primary cilia functions, including the growth signaling.

4) Figure 2G does not allow to compare controls and Twinkle KO. Can the authors compare the data in a single graph?

Au: To visualise the expression level of Foxj1, we compared Foxj1 expression in purified astrocytes to that in the cell suspension. This normalisation is genotype-specific, and thus we prefer to present the data on individual graphs. For the reviewer, we have provided a side-by-side comparison below (Figure A).

Revision Plan

Figure A. *Foxj1* expression in purified astrocytes (ACSA-2⁺ fraction) compared to unsorted cell suspension; RT-qPCR, normalised to *Ywhaz* transcript level. Symbols represent individual preparations ($n = 5$ per genotype). Note that same data are presented in Figure 2G.

5) For promoter analysis, have the authors used all upregulated genes or only those linked to ciliary programs? Is the RFX motif found also in other ISR-driven genes?

Au: We used the top 1000 up-regulated genes from the entire dataset, without choosing selectively cilia-related genes (genes with $q\text{-value} < 0.1$ and $\log_2(\text{FC}) > 0.3$ were selected, and then sorted by $\log_2(\text{FC})$). We carried out the motif enrichment analysis using the CentriMo tool, which is part of the MEME Suite. For clarity, we now highlighted this description with text formatting (page 23, paragraph 3). Additionally, we have carried out the same motif enrichment analysis for the up-regulated ISR^{mt} genes from our dataset. This analysis did not provide any significant enrichment of a motif, most likely due to the low numbers of input genes (only 18 genes in total). We would like to mention here that in the past us and others have shown that ISR^{mt} genes are regulated by e.g. ATF5 (tissues) and ATF3 and 4 (proliferating cells) (Mick et al. 2020; Forsström et al. 2019).

Points raised by Reviewer #2:

1) Data supporting claims on motile cilia

Previous works showed that astrocytes have one single primary cilia. In the current work, the authors observe induction of motile cilia genes. However, they do not address directly, experimentally, whether these results signify a switch between primary and motile mitochondria, or whether this is simply an induction of a dysfunctional ciliogenic program, that thus creates the morphological defects they observe (bent/contorted cilia).

Au: Cilia are classified as primary or motile based on motility property and/or axonemal structure (9+0 and 9+2 microtubular arrangement, respectively), although non-canonical arrangements with unknown motility property were also reported (Odor and Blandau 1985; Gilroy, Singh, and Shahidi 1995). Embryonic nodal cilia are considered to be an exception, as despite possessing a 9+0 axoneme, they are motile. As the reviewer mentions, the majority of differentiated cell types, including astrocytes, are thought to possess a single primary 9+0 cilium. Such cilia are believed to lack protein assemblies required for motility, and thus considered immotile. In turn, motile 9+2 cilia are typically multiple per cell

Revision Plan

and are present on a few specialized cell types which typically face luminal spaces, such as ependymal cells facing ventricular space in the brain.

This paradigm was established during early investigations into ciliary biology using cultured cell lines or on preparations of lumen facing cells, which possess cilia accessible for imaging (for example, 9+2 motile cilia of ependymal cells and airway epithelium, and 9+0 renal primary cilia). **In contrast, investigations into primary cilia motility and microtubular arrangement are lacking for a vast majority of tissue embedded cell types, including astrocytes.** To image microtubular organization with transmission electron microscopy requires a nearly perfect cross-section, which in tissue sections is both very rare and hard to identify, and thus such studies are lacking. Moreover, even studies focused on cultured cells typically capture only part of a cilium, often around ciliary base. Recent advances with serial electron microscopy allowed detailed investigations of primary cilia ultrastructure, surprisingly demonstrating that 9+0 arrangement of primary cilia is present at the cilium base, but differs along the axoneme (Sun et al. 2019; Kiesel et al. 2020). Most recently, it was also shown that cilia of islet cells, classically defined as primary cilia, may be in fact be motile and express motile cilia factors (Cho et al. 2021). Finally, cells of lung epithelia, that possess motile 9+2 cilia, in development and upon injury transiently possess a primary cilium (Jain et al. 2010). These studies emphasize that the diversity of cilia types is more complex than an original and simplistic classification of primary and motile and likely includes cilia with hybrid properties. Moreover, the evidence show that individual cell types can regulate their cilia composition and function during differentiation and disease states.

To shed light onto the question raised by the reviewer, we already went on an extensive endeavor in attempts to investigate ultrastructure of contorted cilia in our model. This included establishing a faithful cell culture system for electron tomography, as well as development of a new approach to use correlative light and electron serial block-face electron microscopy in vivo (described below in more detail). Unfortunately, the resolution was not sufficient to determine ciliary microtubule composition (see below). Within the context of this study, we cannot examine also functional motility of astrocyte cilia due to significant technical challenges. Such approaches exist for lumen-facing cilia but not for tissues where cilia are embedded into tight extracellular space. We acknowledge that the newest focused ion beam scanning electron microscopy (FIB-SEM) applied for large volumes may be a suitable approach to resolve ciliary ultrastructure in finer details, however, the equipment was not available to us. Finally, we have now planned experiments to explore induction of motile cilia components in *TwKO^{astro}* (described above).

In summary, whether the response we observe represents a physiological transition from primary (immotile) to motile cilia or a pathological reaction with consequent changes to cilia morphology is an open question. This is a broad and highly relevant topic that we plan on focusing our future research.

2) In addition, in the discussion they hypothesize three different mechanisms, however this gap is important to provide context for the significance of the results. If indeed there is a switch towards more motile-related functions, this needs to be examined. If it is not possible to measure cilia function in this

Revision Plan

system, the authors should expand their discussion to clearly address the possibly scenarios, and provide functional context, considering the functional effects seen in similar cell culture experiments (Bae et al.), or in other systems where bent/contorted cilia were observed.

Au: We have now incorporated additional discussion about different possible scenarios of ciliary changes in astrocytes and have put it in context relative to other published ciliary abnormalities (page 10 paragraph 2, page 11 paragraphs 1 and 2).

3) Tissue specificity and stress specificity in triggering cilia

Figure S2G is an important figure addressing the generalizability of these results. Importantly, Bae et al. showed that other mitochondrial stresses can also trigger the ciliogenic response. To improve the impact of the current work, the authors should deepen and expand this section, using any other gene expression datasets that are relevant and available (other stresses, knockout of Twinkle in other tissues, different organisms, etc).

Au: We are not completely certain we understand what the reviewer is addressing here. There are no transcriptomics data in Bae et al. In our original submission (Fig S2G), we have already carried out comparative analyses of transcriptomics changes between our dataset and other published datasets acquired either from other astrocyte models or Twinkle knockouts from other cell types.

In addition to heat maps, we now also added stacked bar graphs to Figure S2G visualising fractions of upregulated, downregulated, and unchanged genes from the motile cilia list. We also added these numbers to Supplementary table 3.

4) 1. The last section of the results, discussing metabolic and lipid changes in response to the mitochondrial stress, seems out of place. At the current version, there is no conceptual connection between the cilia data (Figure 1-4) and the metabolic section (supplementary figure 4), for which, data is largely published in a previous work, and re-analyzed. To possibly improve this, the authors may consider either (1) moving Figure S4 to Figure 1 or S1, and re-write the first section of the results. This may be presented as surveying the effects of the mitochondrial dysfunction using both their previously published metabolomic data, along with their new astrocyte-specific RNA-seq. (2) Figure S4 and the accompanying text may be eliminated from the manuscript.

Au: We thank the reviewer for this suggestion in restructuring this part of the results, and we have now moved the results related to lipid metabolism to Figure 1.

5) Some of the methods sections should be expanded, and methods should be explained in more length rather than citing previous works (e.g. astrocyte sorting, FISH) including experimental details (concentrations, dilutions, incubations, etc) to allow reproducibility.

Revision Plan

Au: We have now added these details to methods section.

6) Number of biological repeats needs to be mentioned for each method/experiment.

Au: We now added this information to figure legend for instances where it was missing.

7) Why wasn't GFAP was used as one of the markers for 1A?

Au: To evaluate astrocyte enrichment with our purification methods, we used three genes (Atpb1b2, Aldh1l1, Slc1a3, expression of which is specific to astrocytes in the mouse brain cortex as established with gene expression studies and mouse genetic reporters (Batiuk et al. 2017; Liddelow et al. 2017; Cahoy et al. 2008; Gong et al. 2003). These markers were also used in other published studies for a similar purpose to assess the purity of purified astrocytes and we consider those sufficient (Liddelow et al. 2017; Batiuk et al. 2017). GFAP expression is increased in TwKO^{astro} brain compared to control astrocytes ((Ignatenko et al. 2018, 2020), and in Figure 3A of this study). Thus, we consider that the expression of this marker in purified astrocytes may reflect not only the purity of cell population, but also changes in its expression due to our genetic manipulation.

8) There is some enrichment (~2 fold) in endothelial markers, the authors should explain in the text this caveat.

Au: Cspg4 is indeed enriched in purified astrocyte population, as previously reported (Batiuk et al. 2017). We now included a sentence highlighting this datapoint (page 4, paragraph 3).

6. The authors state "These results provided a proof of principle that the purified fraction represented the astrocyte population with Twnk-KO". Please revise the sentence, as these results "suggest/show/demonstrate" that the purified fraction represents the astrocyte population. This is especially true due to the inter-cellular effects that are known to occur upon mitochondrial stress, which may affect the levels of these genes in other tissues, despite not having the Twnk-KO within the same cells.

Au: We now revised the sentence as suggested by the reviewer (page 5, paragraph 2).

9) Figure 3E reference in the text has a mistake - "Contorted cilia in TwKOastro included S-shaped and corkscrew-like morphologies, and occasionally the long cilia appeared to form several loops (Fig. 3F, bottom panel)". (there is no bottom panel).

Au: We thank the reviewer for pointing out the mistake. The statement should refer to Fig. 3E and is now corrected.

Revision Plan

10) The text relating to Figure S2F "Finally, among factors contributing to primary cilia signaling, the expression of the sonic hedgehog pathway effector Gli1 was induced in astrocytes sorted from TwKOastro mice (Fig. S2F)" seems misplaced in terms of the logic of the paper. Consider moving to an earlier section, or explaining why this is relevant in its current mention location.

Au: We agree with the reviewer that this sentence appears misplaced and now removed it from the text.

11) Authors are missing a reference for another study that examined astrocytes and mitochondrial stress, 10.1016/j.cmet.2020.05.001, which they should cover and cite.

Au: We now added the reference to the introduction section.

Points raised by Reviewer #3:

1) In Figure 3F, the authors should change the pie chart into an easier to read data visualization, like a stacked bar graph. As is, it is difficult to clearly see the differences between groups on a pie chart.

Au: We have now changed pie charts to stacked bar graphs, as suggested by the reviewer.

2) The authors should include an experiment looking at Foxj1 expression levels via RT-qPCR in wild-type astrocytes, not only GFAP-Cre mice.

Au: Figure 2G shows expression levels of Foxj1 measured using RT-qPCR in both control and TwKO^{astro} purified astrocytes compared to unsorted cell suspension derived from the brain.

4. Description of analyses that authors prefer not to carry out

Points raised by Reviewer #2:

1) Finally, the authors rely only on staining ARL13B to identify cilia in astrocytes, however they use additional methods (Figure 4, scanning electron microscopy) for ependymal cells when examining cilia in other cells. Why was SEM not used to verify their results for astrocytes, the main cell type discussed in the manuscript? If this is experimentally possible, this should be done, as it does not rely on protein markers and antibody, which also show some staining of other cellular components in the manuscript (Figure 3).

Au: Scanning electron microscopy (SEM) is suitable for imaging a specimen surface. This method is useful to image for example ventricular surface, where cilia protrude to the lumen (Figure 4B). The method is not suitable for imaging tissue sections, such as cortex.

Revision Plan

Investigating ciliary ultrastructure in a tissue context is not trivial. In fact, we are aware of only one such study in the mouse brain (Matsumoto et al. 2019). These organelles are small (around 200 nm in diameter) and are only one per cell, meaning that statistically most single transmission electron microscopy micrographs do not capture cilia, and capturing a cross-section is even a rarer event. Moreover, on a single plain electron microscopy photograph it is rather impossible to identify such structures with certainty as cilia.

In hope to find our way around the challenge of electron microscopy imaging of cilia in the brain, we established a cell culture system, where we cultured astrocytes in a chemically defined medium that preserves physiological properties of astrocytes (Foo et al. 2011). Imaging of cilia of cultured astrocytes using electron tomography showed that cilia appeared mostly horizontal and internalized inside the cell (Figure B, for the reviewer). Considering that known ciliary functions are attributed to signal transduction via membrane receptors on the extracellular part of the axoneme, we were not convinced that the cultured cell system faithfully represents cilia architecture of astrocytes. Additionally, horizontal positioning of cilia challenges obtaining a cross-section to establish the microtubular arrangement.

Figure B. Ciliation in primary mouse astrocytes purified using ACSA-2 magnetic beads and cultured in chemically defined media. **Left:** immunofluorescence. Red = ARL13B; green = GFAP, blue = DAPI. **Right:** Electron tomography micrograph. Cilia are marked with arrows.

With this, we concluded that to investigate whether contorted cilia of $TwKO^{astro}$ mice acquire structural features of motile cilia would require correlative light and serial electron microscopy. Due to significant methodological challenges, correlative light and serial electron microscopy on tissues are scarce and existing studies are done using large, abundant landmarks (such as blood vessels) rather than a small, single per cell organelle such as cilia (Kremer et al. 2020; Luckner et al. 2018). Despite this, in collaboration with the Electron Microscopy Unit of the University of Helsinki, we took on the challenge and developed a novel challenging workflow for correlative light serial electron microscopy (Figure C).

Specifically, we labelled cilia using in-house designed viral vector (AAV8:GFAP-ARL13B-eGFP), which we delivered intracranially during postnatal Day 0-1. We then euthanised $TwKO^{astro}$ at the age of contorted

Revision Plan

cilia phenotype manifestation (4-5 months), collected vibratome sections, introduced physical marks by applying a focused moving laser beam (branding), imaged regions of interest with confocal microscopy, completed fixation, staining, and embedding for serial electron microscopy, and finally performed serial block-face electron microscopy. We succeeded in imaging the region of interest, however the resulting resolution was not sufficient to resolve ciliary ultrastructure (Figure C). As already mentioned above, we acknowledge that applying newest FIB-SEM machines instead of block-face microscopy may improve the resolution, however this was not available to us.

After these significant but unfortunately inconclusive efforts in trying to image cilia from the brain parenchyma, we have decided not to continue with this line of experimentation for this manuscript. With further tool development and assay optimization from our end this is an area we would like to expand upon in the future.

Revision Plan

Figure C. **Left:** the workflow for correlative light scanning block-face scanning electron microscopy. **Right:** examples of branding marks (top panel) and micrographs from serial block-face scanning electron microscopy (lower panel, TwKO^{astro} brain cortex). Arrow points to a cilium. Boxed area marks cilia on several serial sections. Stars mark structurally abnormal mitochondria in an astrocyte of TwKO^{astro}. Arrowheads point to healthy mitochondria of neighboring cells.

Points raised by Reviewer #3:

1) One thing that would be important if possible is metabolomics. For example, is aspartate/asparagine levels altered that could trigger ISR. Also, whether L-2HG, fumarate or succinate levels ratio to aKG differ, as this ratio can modulate aKG dioxygenases like TETs. Metabolomics with the RNA seq would be helpful. But I realize it could be challenging from in vivo astrocytes.

Au: We agree with the reviewer that if it would be possible to reliably measure metabolite levels from purified Twnk KO astrocytes, it would be very insightful. The gold standard in the field is to extract metabolites from intact tissues or cells using rapid techniques and controlled conditions. In contrast, the process of astrocyte purification takes about four hours, while half lives of many metabolites are much shorter. For example, in isolated mitochondria metabolites are converted in less than a minute (Labajova et al. 2006). Additionally, many metabolites are naturally labile, where the rate of degradation is dependent on many factors, including temperature. Astrocyte purification includes enzymatic digestion at +37 C, mechanical trituration, multiple centrifugations, and exposure to various reagents. Thus, the usefulness of metabolomics is highest from bulk tissues but not purified cells, as for the latter, quick stabilization of the metabolome to infer true levels of metabolites, to our opinion, is not currently possible. Furthermore, there are examples of metabolomics from astrocytes and neurons purified from mouse models with cell-specific knockouts of mitochondrial proteins, which fail to capture expected differences, possibly highlighting methodological limitations (Göbel et al. 2020; Motori et al. 2020). Spatial metabolomics, which could solve the issue, is still of limited accessibility.

Relevant to this point, we have previously published metabolomics analyses from cortical lysates of TwKO^{astro} mice (Ignatenko et al. 2020). Indeed, levels of some of the tricarboxylic acid cycle intermediates are changed in TwKO^{astro}, which could be one of the triggers of ISR. How these metabolite changes relate to ciliary changes and function in astrocytes will be an important area to focus in the future for our group. However, these are long term investigations beyond the scope of this study.

Batiuk, Mykhailo Y., Filip de Vin, Sandra I. Duqué, Chen Li, Takashi Saito, Takaomi Saido, Mark Fiers, T. Grant Belgard, and Matthew G. Holt. 2017. "An Immunoaffinity-Based Method for Isolating Ultrapure Adult Astrocytes Based on ATP1B2 Targeting by the ACSA-2 Antibody." *The Journal of Biological Chemistry* 292 (21): 8874–91.

Cahoy, John D., Ben Emery, Amit Kaushal, Lynette C. Foo, Jennifer L. Zamanian, Karen S. Christopherson, Yi Xing, et al. 2008. "A Transcriptome Database for Astrocytes, Neurons, and Oligodendrocytes: A New Resource for Understanding Brain Development

Revision Plan

- and Function.” *The Journal of Neuroscience: The Official Journal of the Society for Neuroscience* 28 (1): 264–78.
- Cho, Jung Hoon, Zipeng A. Li, Lifei Zhu, Brian D. Muegge, Henry F. Roseman, Toby Utterback, Louis G. Woodhams, Philip V. Bayly, and Jing W. Hughes. 2021. “Islet Primary Cilia Motility Controls Insulin Secretion.” *BioRxiv*. <https://doi.org/10.1101/2021.12.14.472629>.
- Foo, Lynette C., Nicola J. Allen, Eric A. Bushong, P. Britten Ventura, Won-Suk Chung, Lu Zhou, John D. Cahoy, et al. 2011. “Development of a Method for the Purification and Culture of Rodent Astrocytes.” *Neuron* 71 (5): 799–811.
- Forsström, Saara, Christopher B. Jackson, Christopher J. Carroll, Mervi Kuronen, Eija Pirinen, Swagat Pradhan, Anastasiia Marmyleva, et al. 2019. “Fibroblast Growth Factor 21 Drives Dynamics of Local and Systemic Stress Responses in Mitochondrial Myopathy with MtDNA Deletions.” *Cell Metabolism* 30 (6): 1040-1054.e7.
- Gilroy, C., A. Singh, and E. Shahidi. 1995. “Cilia in the Porcine Bile Ductule: Motile or Sensory?” *Histology and Histopathology* 10 (2): 301–4.
- Gong, Shiaoqing, Chen Zheng, Martin L. Doughty, Kasia Losos, Nicholas Didkovsky, Uta B. Schambra, Norma J. Nowak, et al. 2003. “A Gene Expression Atlas of the Central Nervous System Based on Bacterial Artificial Chromosomes.” *Nature* 425 (6961): 917–25.
- Göbel, Jana, Esther Engelhardt, Patric Pelzer, Vignesh Sakthivelu, Hannah M. Jahn, Milica Jevtic, Kat Folz-Donahue, et al. 2020. “Mitochondria-Endoplasmic Reticulum Contacts in Reactive Astrocytes Promote Vascular Remodeling.” *Cell Metabolism* 31 (4): 791-808.e8.
- Ignatenko, Olesia, Dmitri Chilov, Ilse Paetau, Elena de Miguel, Christopher B. Jackson, Gabrielle Capin, Anders Paetau, Mugen Terzioglu, Liliya Euro, and Anu Suomalainen. 2018. “Loss of MtDNA Activates Astrocytes and Leads to Spongiotic Encephalopathy.” *Nature Communications* 9 (1): 70.
- Ignatenko, Olesia, Joni Nikkanen, Alexander Kononov, Nicola Zamboni, Gulayse Ince-Dunn, and Anu Suomalainen. 2020. “Mitochondrial Spongiotic Brain Disease: Astrocytic Stress and Harmful Rapamycin and Ketosis Effect.” *Life Science Alliance* 3 (9). <https://doi.org/10.26508/lsa.202000797>.
- Jain, Raksha, Jiehong Pan, James A. Driscoll, Jeffrey W. Wisner, Tao Huang, Sean P. Gunsten, Yingjian You, and Steven L. Brody. 2010. “Temporal Relationship between Primary and Motile Ciliogenesis in Airway Epithelial Cells.” *American Journal of Respiratory Cell and Molecular Biology* 43 (6): 731–39.
- Kasahara, Kyosuke, Ko Miyoshi, Shinki Murakami, Ikuko Miyazaki, and Masato Asanuma. 2014. “Visualization of Astrocytic Primary Cilia in the Mouse Brain by Immunofluorescent Analysis Using the Cilia Marker Arl13b.” *Acta Medicinæ Okayama* 68 (6): 317–22.
- Khan, Shahzad S., Yuriko Sobu, Herschel S. Dhekne, Francesca Tonelli, Kerry Berndsen, Dario R. Alessi, and Suzanne R. Pfeffer. 2021. “Pathogenic LRRK2 Control of Primary Cilia and Hedgehog Signaling in Neurons and Astrocytes of Mouse Brain.” *ELife* 10 (October). <https://doi.org/10.7554/eLife.67900>.
- Kiesel, Petra, Gonzalo Alvarez Viar, Nikolay Tsoy, Riccardo Maraspini, Alf Honigmann, and Gaia Pigino. 2020. “The Molecular Structure of Primary Cilia Revealed by Cryo-Electron Tomography.” *BioRxiv*. bioRxiv. <https://doi.org/10.1101/2020.03.20.000505>.
- Kremer, A., E. VAN Hamme, J. Bonnardel, P. Borghgraef, C. J. GuÉrin, M. Guilliams, and S. Lippens. 2020. “A Workflow for 3D-CLEM Investigating Liver Tissue.” *Journal of Microscopy*, October. <https://doi.org/10.1111/jmi.12967>.
- Labajova, Anna, Alena Vojtiskova, Pavla Krivakova, Jiri Kofranek, Zdenek Drahota, and Josef Houstek. 2006. “Evaluation of Mitochondrial Membrane Potential Using a Computerized

Revision Plan

- Device with a Tetraphenylphosphonium-Selective Electrode." *Analytical Biochemistry* 353 (1): 37–42.
- Liddelow, Shane A., Kevin A. Guttenplan, Laura E. Clarke, Frederick C. Bennett, Christopher J. Bohlen, Lucas Schirmer, Mariko L. Bennett, et al. 2017. "Neurotoxic Reactive Astrocytes Are Induced by Activated Microglia." *Nature* 541 (7638): 481–87.
- Luckner, Manja, Steffen Burgold, Severin Filser, Maximilian Scheungrab, Yilmaz Niyaz, Eric Hummel, Gerhard Wanner, and Jochen Herms. 2018. "Label-Free 3D-CLEM Using Endogenous Tissue Landmarks." *iScience* 6 (August): 92–101.
- Matsumoto, Mami, Masato Sawada, Diego García-González, Vicente Herranz-Pérez, Takashi Ogino, Huy Bang Nguyen, Truc Quynh Thai, et al. 2019. "Dynamic Changes in Ultrastructure of the Primary Cilium in Migrating Neuroblasts in the Postnatal Brain." *The Journal of Neuroscience: The Official Journal of the Society for Neuroscience* 39 (50): 9967–88.
- Mick, Eran, Denis V. Titov, Owen S. Skinner, Rohit Sharma, Alexis A. Jourdain, and Vamsi K. Mootha. 2020. "Distinct Mitochondrial Defects Trigger the Integrated Stress Response Depending on the Metabolic State of the Cell." *ELife* 9 (May).
<https://doi.org/10.7554/eLife.49178>.
- Moser, Joanna J., Marvin J. Fritzler, and Jerome B. Rattner. 2009. "Primary Ciliogenesis Defects Are Associated with Human Astrocytoma/Glioblastoma Cells." *BMC Cancer* 9 (December): 448.
- Motori, E., I. Atanassov, S. M. V. Kochan, K. Folz-Donahue, V. Sakthivelu, P. Giavalisco, N. Toni, J. Puyal, and N-G Larsson. 2020. "Neuronal Metabolic Rewiring Promotes Resilience to Neurodegeneration Caused by Mitochondrial Dysfunction." *Science Advances* 6 (35): eaba8271.
- Odor, D. L., and R. J. Blandau. 1985. "Observations on the Solitary Cilium of Rabbit Oviductal Epithelium: Its Motility and Ultrastructure." *The American Journal of Anatomy* 174 (4): 437–53.
- Sipos, Éva, Sámuel Komoly, and Péter Ács. 2018. "Quantitative Comparison of Primary Cilia Marker Expression and Length in the Mouse Brain." *Journal of Molecular Neuroscience: MN* 64 (3): 397–409.
- Sterpka, Ashley, and Xuanmao Chen. 2018. "Neuronal and Astrocytic Primary Cilia in the Mature Brain." *Pharmacological Research: The Official Journal of the Italian Pharmacological Society* 137 (November): 114–21.
- Sterpka, Ashley, Juan Yang, Matthew Strobel, Yuxin Zhou, Connor Pauplis, and Xuanmao Chen. 2020. "Diverged Morphology Changes of Astrocytic and Neuronal Primary Cilia under Reactive Insults." *Molecular Brain* 13 (1): 28.
- Sun, Shufeng, Rebecca L. Fisher, Samuel S. Bowser, Brian T. Pentecost, and Haixin Sui. 2019. "Three-Dimensional Architecture of Epithelial Primary Cilia." *Proceedings of the National Academy of Sciences of the United States of America* 116 (19): 9370–79.

April 19, 2022

Re: JCB manuscript #202203019T

Prof. Anu Suomalainen-Wartiovaara
University of Helsinki
Stem Cells and Metabolism Research Program
Biomedicum Helsinki
Haartmaninkatu 8
Helsinki 00290
Finland

Dear Prof. Suomalainen-Wartiovaara,

Thank you for submitting your manuscript "Mitochondrial dysfunction compromises ciliary homeostasis in astrocytes" from Review Commons. Having assessed your manuscript, the Review Commons reports, and your revision plan we felt the work was potentially suitable as a Report in JCB. We have now also received a report from an additional reviewer with expertise in cilia, which is appended to this letter.

You will see that reviewer #4 also feels that your work is well done and provides interesting and significant new insights. This reviewer's only concern is that the conclusion that ependymal cilia are not affected by the astrocyte specific Twinkle KO is not well supported by data and so asks for additional assays with various ciliary markers in both astrocytes and ependymal cells. We therefore invite you to submit a revision as outlined in your response and also ask that you please address reviewer #4's comments in full.

GENERAL GUIDELINES:

Text limits: Character count for a Report is < 20,000, not including spaces. Please let us know if you will need more space. Count includes title page, abstract, introduction, the joint Results & Discussion, and acknowledgments. Count does not include materials and methods, figure legends, references, tables, or supplemental legends.

Figures: Reports may have up to 5 main text figures. To avoid delays in production, figures must be prepared according to the policies outlined in our Instructions to Authors, under Data Presentation, <https://jcb.rupress.org/site/misc/ifora.xhtml>. All figures in accepted manuscripts will be screened prior to publication.

*****IMPORTANT:** It is JCB policy that if requested, original data images must be made available. Failure to provide original images upon request will result in unavoidable delays in publication. Please ensure that you have access to all original microscopy and blot data images before submitting your revision. ***

Supplemental information: There are strict limits on the allowable amount of supplemental data. Reports may have up to 3 supplemental figures. Up to 10 supplemental videos or flash animations are allowed. A summary of all supplemental material should appear at the end of the Materials and methods section.

Please note that JCB now requires authors to submit Source Data used to generate figures containing gels and Western blots with all revised manuscripts. This Source Data consists of fully uncropped and unprocessed images for each gel/blot displayed in the main and supplemental figures. If your paper will include cropped gel and/or blot images, please be sure to provide one Source Data file for each figure that contains gels and/or blots along with your revised manuscript files. File names for Source Data figures should be alphanumeric without any spaces or special characters (i.e., SourceDataF#, where F# refers to the associated main figure number or SourceDataFS# for those associated with Supplementary figures). The lanes of the gels/blots should be labeled as they are in the associated figure, the place where cropping was applied should be marked (with a box), and molecular weight/size standards should be labeled wherever possible. Source Data files will be made available to reviewers during evaluation of revised manuscripts and, if your paper is eventually published in JCB, the files will be directly linked to specific figures in the published article.

The typical timeframe for revisions is three to four months. While most universities and institutes have reopened labs and

allowed researchers to begin working at nearly pre-pandemic levels, we at JCB realize that the lingering effects of the COVID-19 pandemic may still be impacting some aspects of your work, including the acquisition of equipment and reagents. Therefore, if you anticipate any difficulties in meeting this aforementioned revision time limit, please contact us and we can work with you to find an appropriate time frame for resubmission. Please note that papers are generally considered through only one revision cycle, so any revised manuscript will likely be either accepted or rejected.

Thank you for this interesting contribution to Journal of Cell Biology. You can contact us at the journal office with any questions, cellbio@rockefeller.edu or call (212) 327-8588.

Sincerely,

Richard Youle, PhD
Monitoring Editor
Journal of Cell Biology

Dan Simon, PhD
Scientific Editor
Journal of Cell Biology

Reviewer #4 (Comments to the Authors (Required)):

The manuscript by Ignatenko et al shows that conditional knockout of the mitochondrial DNA helicase *twinkle* induces an multiciliogenesis program in astrocytes, which have a single primary cilium. This motile ciliogenesis program was activated by blocking EGFR signaling in cultured astrocytes, but not in heart-specific TwKO, reactive astrocytes, or aged astrocytes, demonstrating cell-type specificity. In TwKOastro mice, astrocyte cilia are longer and misshapen, while ependymal multiciliated cells appeared superficial normal. This manuscript is extremely well written and a delight to read, the data - for the most part - are convincing, and this new and surprising connection between mitochondria and cilia will appeal to the broad readership of the JCB.

I agree with the assessments from Review Commons, and the Authors' proposed revision plan. It is critical to look at additional ciliary markers in both astrocytes and ependymal cells. For the latter, I do not think authors can conclude "Ependymal cilia do not show apparent abnormalities upon mitochondrial dysfunction" based on the immunofluorescent and SEM images shown in Figure 4B. To my eye, the cilia appear possibly misoriented in TwKOastro animals. At a minimum, authors should use α -Acetylated tubulin for axonemes and α - γ tubulin for basal bodies (this antibody in conjunction with α -beta-catenin would also be useful for examining morphology and polarity of ependymal cilia basal bodies)

Cilia consume ATP yet have no mitochondria. There is a growing body of literature related to ciliary metabolic demands and energy sources, particularly in the kidney and kidney diseases. Nephron development is one of the upregulated pathways in TwKOastro (Fig 2a). If discussion space were unlimited, it would be interesting to hear authors' thoughts on this potential connection to their work.

This review on cilia in the brain was published while this manuscript was being revised. Tereshko et al write "In contrast to neurons, the role of cilia on mature astrocytes is poorly understood." This work by Ignatenko et al is important from fundamental cell biology and human disease (neurodegeneration and ciliopathies) perspectives.

Primary cilia in the postnatal brain: Subcellular compartments for organizing neuromodulatory signaling. Tereshko L, Turrigiano GG, Sengupta P. *Curr Opin Neurobiol.* 2022 Apr 8;74:102533. doi: 10.1016/j.conb.2022.102533. Online ahead of print.

Revision Plan

Manuscript number: RC-2022-01235

Corresponding author(s): Gulayse, Ince Dunn; Anu, Suomalainen

1. General Statements [optional]

Au: We thank for the highly favourable and enthusiastic comments of the three reviewers, and their highly supportive assessment of our work. All of them appreciated the conceptual novelty of our study. They suggested minor improvements, which we have addressed point-by-point below.

We have previously shown that mitochondrial spongiotic encephalopathy, a common and devastating manifestation of patients with mitochondrial brain disease, can be caused by pathologically reactivated astrocytes. Here, we report novel molecular mechanisms of astrocyte reactivation as a consequence of mitochondrial dysfunction, specifically, mtDNA depletion. Our transcriptomics analyses from purified cortical astrocytes led to the identification of an unexpected and novel pathway: a wide activation of motile ciliogenesis program in astrocytes that are known to harbor single primary cilium. We further carried out detailed functional and morphological experiments *in vivo* to reveal that an aberrant upregulation of a motile ciliogenesis program was associated with primary cilia elongation and abnormal morphology, indicating that mitochondrial brain diseases have a component of metabolic ciliopathy.

Update: we thank also the fourth reviewer recruited by JCB after the review at Review Commons platform for the favorable assessment of our work and respond to their comments below.

Reviewer #1 (Evidence, reproducibility and clarity (Required)):

This well-written and convincing manuscript surprisingly links for the first time mitochondrial dysfunction to cilia homeostasis in a specific cell type of the brain, the astrocyte. Previously the authors have beautifully demonstrated that deletion of the Twinkle helicase in astrocytes leads to a more severe phenotype than the deletion of the gene in neurons, resembling human spongiotic encephalopathy. Furthermore, they showed that mitochondrial dysfunction elicited in astrocytes an integrated stress response characterized by specific features. Now they go a step further, by purifying astrocytes deleted of Twinkle and analyzing their global transcriptional profile. Surprisingly, they detect an upregulation of genes linked to ciliary function, especially to the motile cilium. This response is caused by the activation of the FOXJ1 and RFX transcription factors, which are known master regulators of motile ciliogenesis. Strikingly, cilia of Twinkle-deficient astrocytes appeared abnormal, being elongated or with contorted morphology. This phenotype was not observed in ependymal cells that are also targeted by the Cre line used by the authors, indicating cell specificity. The authors propose that this response is an astrocyte-specific anabolic response due to the ISR activation.

Overall, this is an original study that opens a new paradigm: the possible implication of abnormal ciliogenesis as a cell-specific response to mitochondrial dysfunction. The limitation of this study, as the authors openly discuss, is that it is unclear what is the pathological significance of this abnormal ciliary

Revision Plan

program. Unravelling the functional significance of this pathway will however require additional years of study.

Au: We thank the reviewer for the positive evaluation of our work and thoughtful summary of findings.

Reviewer #1 (Significance (Required)):

An important aspect explaining the specificity of the disease phenotype in case of mitochondrial dysfunction is the tissue-specificity of elicited stress response pathways. This manuscript provides an entirely novel point of view, implicating abnormal ciliogenesis as a novel target of this response specifically in astrocytes, which are important players in the pathogenesis of mitochondrial brain disease. In the context of several papers now showing stress responses upon mitochondrial dysfunction, this paper highlights a novel and totally unexpected pathway. I expect that the audience for this manuscript will be far beyond scientists working in the field of mitochondrial dysfunction and neurodegeneration, like this reviewer. I expect that these findings will also appeal to the currently rather separate field of cilia function and brain disease.

Au: We thank the reviewer for enthusiastic reading of our work and seeing potential in its impact.

Reviewer #2 (Evidence, reproducibility and clarity (Required)):

In this manuscript by Ignatenko et al., the authors follow up on two previous works from the same authors, using mice with a defect in mitochondrial function specifically in astrocytes. While works in recent years interrogated the effects of knocking out mitochondrial genes mainly in peripheral tissues (e.g. metabolic organs), the current work focuses on astrocytes. These non-neuronal cells of the nervous system have been less studied in the context of mitochondrial defects. In the current work, the authors focus on exploring the transcriptional response in astrocytes to the mitochondrial insult, which revealed induction of cilia-related processes, similar to the findings of Bae et al 2019 (10.1038/s41419-019-2184-y) in cultured cells. This work provides conceptual insight and advancement at the level of cell biology in an in-vivo model, illustrating a morphological and transcriptional change in astrocytes upon mitochondrial stress. The work described by Ignatenko et al. will be interesting to the field of mitochondrial biology, and researchers in cell biology, as it shows that the ciliogenic changes occur also in an in-vivo model.

The authors' work is comprehensive, rigorous, and elegant. Experiments are well-controlled and the statements in the manuscript are generally convincing considering the data presented in the figures. The manuscript is beautifully written and easy to follow.

Au: We thank the reviewer for such favorable assessment our work.

Reviewer #2 (Significance (Required)):

See comments above.

Reviewer #3 (Evidence, reproducibility and clarity (Required)):

In this paper, Ignatenko, et al. describe a set of experiments demonstrating the link between mitochondrial dysfunction in astrocytes and motile ciliogenesis. Using astrocytes isolated from

Revision Plan

TwKOastro mice, which possess a knockout of the mitochondrial DNA helicase Twinkle in astrocytes leading to reactive gliosis, the researchers provide results from several experiments to support their central claim that respiratory chain deficiency in astrocytes activates FOXJ1 and RFX transcription factors which results in changes to motile ciliogenesis. Using RNASeq, the researchers demonstrated mtDNA mutations impact motile cilia in astrocytes taken from TwKOastro mice. Previously reported ChIPSeq data from ependymal cells was then used to determine which transcription factor sequences were enriched in TwKOastro mice. Foxj1 expression was observed in astrocytes, leading researchers to hypothesize that mitochondrial damage in astrocytes caused by the deletion of Twinkle leads to an increase in Foxj1 expression, producing an increase in motile ciliogenic program in astrocytes. Scanning electron microscopy demonstrated that the primary cilium of TwKOastro mice is deformed compared to control. Additionally, the researchers demonstrated that this response to mitochondrial dysfunction is cell-type specific to astrocytes by comparing the results from the above experiments to other models of mitochondrial dysfunction in other cell types. These findings contribute more information to the field of astrocyte involvement in the pathogenesis of neurodegenerative disorders. Additionally, these experiments demonstrate previously unknown roles of RFX and FOXJ1 in adult astrocytes and in integrated stress response in the mitochondria. Paper is well done and properly interpreted! It is also one of the first to mechanistically link mitochondrial function to astrocytes.

Au: We thank the reviewer for the endorsement of our work.

Reviewer #3 (Significance (Required)):

There is very little known about how mitochondrial function controls astrocytes. paper is one of the first!

Au: We share the excitement expressed by the reviewer! Thank you!

2. Description of the planned revisions

Points raised by Reviewer #2:

1) While the researchers did a terrific job, there are two points that could be improved. First - the main point, that mitochondrial dysfunction triggers changes in cilium-related processes, is mostly supported by gene expression changes, and the validation experiment relies solely on the pan-cilia marker ARL13B. The authors show that a transcriptional program of motile cilia components is induced, however this was not experimentally verified beyond RNA levels (see below).

Finally, the only change observed in cilia upon the mitochondrial defect is morphological, with no clear functional outcomes.

Au: Previous published evidence for astrocyte cilia analyses are based entirely on ARL13B as a marker (Sterpka and Chen 2018; Kasahara et al. 2014; Sipos, Komoly, and Ács 2018; Khan et al. 2021). Another commonly used marker to visualize axonemal morphology is AC3, which in the brain displays higher

Revision Plan

signal intensity in neurons compared to astrocytes, and thus is not used for visualization of cilia in astrocytes (Sipos, Komoly, and Ács 2018; Sterpka and Chen 2018); our unpublished observations).

We agree with the reviewer that our analysis of motile ciliogenesis program is based on RNA level changes. To investigate protein level changes in TwKO^{astro} brain, we previously attempted immunostainings for selected proteins. We have now ordered a panel of antibodies against motile cilia components to test these further. Single cilia are very small structures that are difficult to image from specific cell types in the 3D space of the brain, but we certainly hope that this experiment will shed information about protein level changes in the astrocyte cilia of our mouse model.

Finally, the functional consequences of ciliary pathology in astrocytes and changes to ciliary signaling pathways form a whole new area of research that we are very much interested in pursuing. However, this is a completely new line of experimentation involving generating new mouse lines, pharmacological interventions, and not in the scope of this article.

Update (new experiments and data in bold): To respond to the reviewer's comment, to test for upregulation of the motile cilia program at the protein level, we now performed *i) proteomics analysis of purified cortical astrocytes of adult mice (new Fig. 3D and **Suppl. tables 3 and 6**) and ii) a series of immunofluorescence stainings of selected motile cilia markers (new Figures 4G-H and S2F-G).*

- i) We pooled four cerebral cortices from TwKO^{astro} and Ctrl mice per preparation and carried out proteomics analysis using data independent acquisition LC-MS/MS with QExactive HF mass spectrometer. In total, we detected 21 ciliary proteins (11 motile cilia-specific and 10 primary cilia or pan-ciliary proteins) in our proteomics dataset (**new Fig. 3D and Suppl. tables 3 and 6**). Our analysis demonstrated the induction of six motile cilia specific components at the protein level (six out of 11 detected motile ciliary proteins). Additionally, the analysis also showed induction of an five pan-ciliary intraflagellar transport components, which are essential for ciliary formation and maintenance and implicated into control of ciliary length as described by (Broekhuis, Leong, and Jansen 2013; Avasthi and Marshall 2012; Ishikawa and Marshall 2017). The low coverage of the ciliome relatively to our RNAseq dataset is likely related to the generally low abundance of ciliary proteins, as primary cilia are small singular structures per cell.

- ii) We tested a panel of antibodies against the motile cilia factors upregulated in our RNAseq dataset from purified mouse astrocytes (anti-GAS8, RSPH4a, DRC3, SPAG6, DNAH5). As a positive control, we imaged ependymal cells that possess motile cilia and stained with the antibody against the well-described cilia component ARL13B. Antibodies against GAS8 and RSPH4a showed a specific signal (*new Fig. S2F*). GAS8 showed a prominent increase in cytoplasmic signal in cortical astrocytes of TwKO^{astro}, while RSPH4A showed no induction (**new Figs. 4G-H and S2G**). Despite our extensive staining optimization efforts, other antibodies did not show signal in the brain. Those antibodies were originally validated using cell types where motile cilia factors are highly expressed, such as human sperm cells, but not

Revision Plan

for mouse brain parenchyma.

Taken together, we now provide evidence with our proteomics results that in addition to a robust upregulation at the RNA level, a number of motile cilia components are also up-regulated at the protein level in TwKO^{astro} cerebral cortex. Currently, we were able to provide additional spatial resolution with immunofluorescence only for GAS8, but will continue to investigate this topic in future projects. In addition, our in situ RNAScope hybridization data show that at the RNA level, *Foxj1* TF that regulates motile ciliogenesis, is up-regulated in the brain parenchyma (Fig. 2H). We thank the reviewer for endorsing the endeavor as the new data are an important addition to our study.

2) The authors follow gene-expression changes, but it is not clear whether the components they see induced in fact get incorporated into the cilia (this could be examined, for example, for genes identified in their dataset using antibodies).

Au: To address this, we will carry out a series of immunostainings on the brains of control and TwKO^{astro} mice using commercially available antibodies against motile cilia factors, which are induced in our dataset (please see more detailed explanation above).

Update: As indicated above, we now added proteomics data on astrocytes sorted from TwKO^{astro} mice (new Fig. 3D and Suppl. tables 3 and 6) and anti-GAS8 and anti-RSPH4a stainings to the manuscript (new Figs. 4G-H and S2F-G). GAS8 is prominently induced in the cytoplasm and does not show incorporation into cilia, which can be related to impaired trafficking into cilia. ARL13B appears to incorporate into cilia of cortical TwKO^{astro} normally (both endogenous protein and from a viral AAV vector, data not shown) and structure of the axoneme is maintained, which indicate at least partial preservation of trafficking into the cilium. Thus, it is possible that the aberrantly induced motile cilia components fail to integrate into the existing primary cilium of astrocytes. In turn, their accumulation in the cytoplasm may lead to secondary consequences for cell fitness and homeostasis. As indicated in our response above, our extensive testing indicates that most commercially available antibodies do not detect by immunofluorescence other motile cilia components in vivo in the brain parenchyma, which is consistent with the lack of such data in previous publications.

1. Description of the revisions that have already been incorporated in the transferred manuscript

Points raised by Reviewer #1:

1) The quantification of the beautiful ciliary phenotype is a crucial result of this study. More explanation should be provided in the material and methods about the method used to quantify the length of the cilium. It seems difficult to quantify cilia with contorted morphology. The authors should also specify if the analyses were conducted blind to the genotype.

Revision Plan

Au: We added details of the cilia length analysis to methods section. As a proxy of ciliary length, we used the longest principal axis of a minimal rectangular box, which encloses a cilium on confocal microscopy images. We agree with the reviewer that it is difficult to estimate the exact length of cilia with contorted morphology. For example, if a cilium is contorted and makes loops, the length of such cilia would be underestimated. The majority of cilia in our datasets are however straight or bent (Figure 2F), and thus the method of choice is adequate for the purpose to investigate the distribution of lengths. Cilia analysis was done blinded to the genotype, which we now indicated in the Methods section.

Update: Figure 2F is now Figure 4F

2) The lipidomic data are certainly interesting, but it is not clear why the authors decide to selectively show these data (from the dataset described in the previous Ignanenko et al., 2020 paper). The link to the previous part of the manuscript and the relevance for the abnormal cilia formation is not clear.

Also the statement that the abnormal lipid profiles depend on alteration in beta-oxidation is just a speculation. This can be more clear in the discussion.

Au: We agree with the reviewer that changes to lipid metabolism and cilia maintenance at first sight do not seem to be connected. However, we believe that presenting both up-regulated and down-regulated genes in this manuscript provides a more holistic view to the readers. We chose to present the lipidomics data from our previous study so that the readers are presented side-by-side the astrocyte-specific gene expression changes and biochemical changes relating to lipid metabolism from TwKO^{astro} mice (unpublished data). The analysis of lipid metabolite levels is based on a dataset we previously published.

In the updated version of our manuscript, we have provided a general introduction to our RNAseq results, including the lipid metabolism findings, in Figure 1, and have focused exclusively on the cilia findings for the remainder of the manuscript.

We also modified our statement about beta oxidation to highlight that it is our hypothesis.

3) Figure S2G: The authors could discuss why cultured astrocytes deprived of HBEGF or treated with EGFRi show a similar response as observed here. Is it possible to find some parallelism, for example in the metabolic response?

Au: We agree that this is an interesting observation and we have now added a sentence highlighting the potential existence of overlapping molecular changes in mtDNA-depleted and EGF signaling-inhibited astrocytes (page 8, paragraph 1). Currently, we do not know whether there are overlapping mechanisms and what these might be in these two different means of perturbing astrocyte physiology. We would like to mention that some other studies of e.g. mouse models of familial Parkinson's disease and traumatic brain injury, have reported changes to cilia morphology (Khan et al. 2021; Sterpka et al. 2020; Moser,

Revision Plan

Fritzler, and Rattner 2009) . The lack of transcriptomics data in these studies excluded them from our comparative analysis in Fig S2G. It will certainly be fascinating to uncover how and why astrocytes respond to specific insults (for example, by restructuring ciliary morphology or having impaired turnover of ciliary factors), and how this impacts the main primary cilia functions, including the growth signaling.

Update: Figure S2G is now Figure 3E.

4) Figure 2G does not allow to compare controls and Twinkle KO. Can the authors compare the data in a single graph?

Au: To visualise the expression level of *Foxj1*, we compared *Foxj1* expression in purified astrocytes to that in the cell suspension. This normalisation is genotype-specific, and thus we prefer to present the data on individual graphs. For the reviewer, we have provided a side-by-side comparison below (Figure A).

Figure A. *Foxj1* expression in purified astrocytes (ACSA-2⁺ fraction) compared to unsorted cell suspension; RT-qPCR, normalised to *Ywhaz* transcript level. Symbols represent individual preparations (n = 5 per genotype). Note that same data are presented in Figure 2G.

5) For promoter analysis, have the authors used all upregulated genes or only those linked to ciliary programs? Is the RFX motif found also in other ISR-driven genes?

Au: We used the top 1000 up-regulated genes from the entire dataset, without choosing selectively cilia-related genes (genes with q-value < 0.1 and log₂(FC) > 0.3 were selected, and then sorted by log₂(FC)). We carried out the motif enrichment analysis using the CentriMo tool, which is part of the MEME Suite. For clarity, we now highlighted this description with text formatting (page 23, paragraph 2). Additionally, we have carried out the same motif enrichment analysis for the up-regulated ISR^{mt} genes from our dataset. This analysis did not provide any significant enrichment of a motif, most likely due to the low numbers of input genes (only 18 genes in total). We would like to mention here that in the past us and others have shown that ISR^{mt} genes are regulated by e.g. ATF5 (tissues) and ATF3 and 4 (proliferating cells) (Mick et al. 2020; Forsström et al. 2019).

Points raised by Reviewer #2:

Revision Plan

1) Data supporting claims on motile cilia

Previous works showed that astrocytes have one single primary cilia. In the current work, the authors observe induction of motile cilia genes. However, they do not address directly, experimentally, whether these results signify a switch between primary and motile mitochondria, or whether this is simply an induction of a dysfunctional ciliogenic program, that thus creates the morphological defects they observe (bent/contorted cilia).

Au: Cilia are classified as primary or motile based on motility property and/or axonemal structure (9+0 and 9+2 microtubular arrangement, respectively), although non-canonical arrangements with unknown motility property were also reported (Odor and Blandau 1985; Gilroy, Singh, and Shahidi 1995). Embryonic nodal cilia are considered to be an exception, as despite possessing a 9+0 axoneme, they are motile. As the reviewer mentions, the majority of differentiated cell types, including astrocytes, are thought to possess a single primary 9+0 cilium. Such cilia are believed to lack protein assemblies required for motility, and thus considered immotile. In turn, motile 9+2 cilia are typically multiple per cell and are present on a few specialized cell types which typically face luminal spaces, such as ependymal cells facing ventricular space in the brain.

This paradigm was established during early investigations into ciliary biology using cultured cell lines or on preparations of lumen facing cells, which possess cilia accessible for imaging (for example, 9+2 motile cilia of ependymal cells and airway epithelium, and 9+0 renal primary cilia). *In contrast, investigations into primary cilia motility and microtubular arrangement are lacking for a vast majority of tissue embedded cell types, including astrocytes.* To image microtubular organization with transmission electron microscopy requires a nearly perfect cross-section, which in tissue sections is both very rare and hard to identify, and thus such studies are lacking. Moreover, even studies focused on cultured cells typically capture only part of a cilium, often around ciliary base. Recent advances with serial electron microscopy allowed detailed investigations of primary cilia ultrastructure, surprisingly demonstrating that 9+0 arrangement of primary cilia is present at the cilium base, but differs along the axoneme (Sun et al. 2019; Kiesel et al. 2020). Most recently, it was also shown that cilia of islet cells, classically defined as primary cilia, may be in fact be motile and express motile cilia factors (Cho et al. 2021). Finally, cells of lung epithelia, that possess motile 9+2 cilia, in development and upon injury transiently possess a primary cilium (Jain et al. 2010). These studies emphasize that the diversity of cilia types is more complex than an original and simplistic classification of primary and motile and likely includes cilia with hybrid properties. Moreover, the evidence show that individual cell types can regulate their cilia composition and function during differentiation and disease states.

To shed light onto the question raised by the reviewer, we already went on an extensive endeavor in attempts to investigate ultrastructure of contorted cilia in our model. This included establishing a faithful cell culture system for electron tomography, as well as development of a new approach to use correlative light and electron serial block-face electron microscopy *in vivo* (described below in more detail). Unfortunately, the resolution was not sufficient to determine ciliary microtubule composition (see below). Within the context of this study, we cannot examine also functional motility of astrocyte cilia due

Revision Plan

to significant technical challenges. Such approaches exist for lumen-facing cilia but not for tissues where cilia are embedded into tight extracellular space. We acknowledge that the newest focused ion beam scanning electron microscopy (FIB-SEM) applied for large volumes may be a suitable approach to resolve ciliary ultrastructure in finer details, however, the equipment was not available to us. Finally, we have now planned experiments to explore induction of motile cilia components in TwKO^{astro} (described above).

In summary, whether the response we observe represents a physiological transition from primary (immotile) to motile cilia or a pathological reaction with consequent changes to cilia morphology is an open question. This is a broad and highly relevant topic that we plan on focusing our future research.

Update: As indicated above, we now added anti-GAS8 and anti-RSPH4a immunofluorescence stainings to the manuscript (**new Figs. 4G-H and S2F-G**). GAS8 is induced in the cytoplasm of cortical astrocytes in TwKO^{astro} mice.

2) In addition, in the discussion they hypothesize three different mechanisms, however this gap is important to provide context for the significance of the results. If indeed there is a switch towards more motile-related functions, this needs to be examined. If it is not possible to measure cilia function in this system, the authors should expand their discussion to clearly address the possibly scenarios, and provide functional context, considering the functional effects seen in similar cell culture experiments (Bae et al.), or in other systems where bent/contorted cilia were observed.

Au: We have now incorporated additional discussion about different possible scenarios of ciliary changes in astrocytes and have put it in context relative to other published ciliary abnormalities (page 9-10).

3) Tissue specificity and stress specificity in triggering cilia

Figure S2G is an important figure addressing the generalizability of these results. Importantly, Bae et al. showed that other mitochondrial stresses can also trigger the ciliogenic response. To improve the impact of the current work, the authors should deepen and expand this section, using any other gene expression datasets that are relevant and available (other stresses, knockout of Twinkle in other tissues, different organisms, etc).

Au: We are not completely certain we understand what the reviewer is addressing here. There are no transcriptomics data in Bae et al. In our original submission (Fig S2G), we have already carried out comparative analyses of transcriptomics changes between our dataset and other published datasets acquired either from other astrocyte models or Twinkle knockouts from other cell types.

In addition to heat maps, we now also added stacked bar graphs to Figure S2G visualising fractions of upregulated, downregulated, and unchanged genes from the motile cilia list. We also added these numbers to Supplementary table 3.

Revision Plan

Update: Figure S2G is now Figure 3E.

4) 1. The last section of the results, discussing metabolic and lipid changes in response to the mitochondrial stress, seems out of place. At the current version, there is no conceptual connection between the cilia data (Figure 1-4) and the metabolic section (supplementary figure 4), for which, data is largely published in a previous work, and re-analyzed. To possibly improve this, the authors may consider either (1) moving Figure S4 to Figure 1 or S1, and re-write the first section of the results. This may be presented as surveying the effects of the mitochondrial dysfunction using both their previously published metabolomic data, along with their new astrocyte-specific RNA-seq. (2) Figure S4 and the accompanying text may be eliminated from the manuscript.

Au: We thank the reviewer for this suggestion in restructuring this part of the results, and we have now moved the results related to lipid metabolism to Figure 1.

5) Some of the methods sections should be expanded, and methods should be explained in more length rather than citing previous works (e.g. astrocyte sorting, FISH) including experimental details (concentrations, dilutions, incubations, etc) to allow reproducibility.

Au: We have now added these details to methods section.

6) Number of biological repeats needs to be mentioned for each method/experiment.

Au: We now added this information to figure legend for instances where it was missing.

7) Why wasn't GFAP was used as one of the markers for 1A?

Au: To evaluate astrocyte enrichment with our purification methods, we used three genes (Atp1b2, Aldh1l1, Slc1a3, expression of which is specific to astrocytes in the mouse brain cortex as established with gene expression studies and mouse genetic reporters (Batiuk et al. 2017; Liddelw et al. 2017; Cahoy et al. 2008; Gong et al. 2003). These markers were also used in other published studies for a similar purpose to assess the purity of purified astrocytes and we consider those sufficient (Liddelw et al. 2017; Batiuk et al. 2017). GFAP expression is increased in TwKO^{astro} brain compared to control astrocytes (Ignatenko et al. 2018, 2020), and in Figure 3A of this study). Thus, we consider that the expression of this marker in purified astrocytes may reflect not only the purity of cell population, but also changes in its expression due to our genetic manipulation.

Update: Figure 3A is now Figure 4A.

8) There is some enrichment (~2 fold) in endothelial markers, the authors should explain in the text this caveat.

Revision Plan

Au: *Cspg4* is indeed enriched in purified astrocyte population, as previously reported (Batiuk et al. 2017). We now included a sentence highlighting this datapoint (page 4, paragraph 1).

6. The authors state "These results provided a proof of principle that the purified fraction represented the astrocyte population with *Twnk*-KO". Please revise the sentence, as these results "suggest/show/demonstrate" that the purified fraction represents the astrocyte population. This is especially true due to the inter-cellular effects that are known to occur upon mitochondrial stress, which may affect the levels of these genes in other tissues, despite not having the *Twnk*-KO within the same cells.

Au: We now revised the sentence as suggested by the reviewer (page 5, paragraph 2).

9) Figure 3E reference in the text has a mistake - "Contorted cilia in *Twk*Oastro included S-shaped and corkscrew-like morphologies, and occasionally the long cilia appeared to form several loops (Fig. 3F, bottom panel)". (there is no bottom panel).

Au: We thank the reviewer for pointing out the mistake. The statement should refer to Fig. 3E and is now corrected.

Update: Figure 3 is Figure 4 now.

10) The text relating to Figure S2F "Finally, among factors contributing to primary cilia signaling, the expression of the sonic hedgehog pathway effector *Gli1* was induced in astrocytes sorted from *Twk*Oastro mice (Fig. S2F)" seems misplaced in terms of the logic of the paper. Consider moving to an earlier section, or explaining why this is relevant in its current mention location.

Au: We agree with the reviewer that this sentence appears misplaced and now removed it from the text.

11) Authors are missing a reference for another study that examined astrocytes and mitochondrial stress, 10.1016/j.cmet.2020.05.001, which they should cover and cite.

Au: We now added the reference to the introduction section.

Points raised by Reviewer #3:

1) In Figure 3F, the authors should change the pie chart into an easier to read data visualization, like a stacked bar graph. As is, it is difficult to clearly see the differences between groups on a pie chart.

Au: We have now changed pie charts to stacked bar graphs, as suggested by the reviewer.

Update: Figure 3F is Figure 4F now.

Revision Plan

2) The authors should include an experiment looking at Foxj1 expression levels via RT-qPCR in wild-type astrocytes, not only GFAP-Cre mice.

Au: Figure 2G shows expression levels of Foxj1 measured using RT-qPCR in both control and TwKO^{astro} purified astrocytes compared to unsorted cell suspension derived from the brain.

3. Description of analyses that authors prefer not to carry out

Points raised by Reviewer #2:

1) Finally, the authors rely only on staining ARL13B to identify cilia in astrocytes, however they use additional methods (Figure 4, scanning electron microscopy) for ependymal cells when examining cilia in other cells. Why was SEM not used to verify their results for astrocytes, the main cell type discussed in the manuscript? If this is experimentally possible, this should be done, as it does not rely on protein markers and antibody, which also show some staining of other cellular components in the manuscript (Figure 3).

Au: Scanning electron microscopy (SEM) is suitable for imaging a specimen surface. This method is useful to image for example ventricular surface, where cilia protrude to the lumen (Figure 4B). The method is not suitable for imaging tissue sections, such as cortex.

Investigating ciliary ultrastructure in a tissue context is not trivial. In fact, we are aware of only one such study in the mouse brain (Matsumoto et al. 2019). These organelles are small (around 200 nm in diameter) and are only one per cell, meaning that statistically most single transmission electron microscopy micrograph do not capture cilia, and capturing a cross-section is even a rarer event. Moreover, on a single plain electron microscopy photographs it is rather impossible to identify such structures with certainty as cilia.

In hope to find our way around the challenge of electron microscopy imaging of cilia in the brain, we established a cell culture system, where we cultured astrocytes in a chemically defined medium that preserves physiological properties of astrocytes (Foo et al. 2011). Imaging of cilia of cultured astrocytes using electron tomography showed that cilia appeared mostly horizontal and internalized inside the cell (Figure B, for the reviewer). Considering that known ciliary functions are attributed to signal transduction via membrane receptors on the extracellular part of the axoneme, we were not convinced that the cultured cell system faithfully represents cilia architecture of astrocytes. Additionally, horizontal positioning of cilia challenges obtaining a cross-section to establish the microtubular arrangement.

Revision Plan

Figure B. Ciliation in primary mouse astrocytes purified using ACSA-2 magnetic beads and cultured in chemically defined media. *Left*: immunofluorescence. Red = ARL13B; green = GFAP, blue = DAPI. *Right*: Electron tomography micrograph. Cilia are marked with arrows.

With this, we concluded that to investigate whether contorted cilia of TwKO^{astro} mice acquire structural features of motile cilia would require correlative light and serial electron microscopy. Due to significant methodological challenges, correlative light and serial electron microscopy on tissues are scarce and existing studies are done using large, abundant landmarks (such as blood vessels) rather than a small, single per cell organelle such as cilia (Kremer et al. 2020; Luckner et al. 2018). Despite this, in collaboration with the Electron Microscopy Unit of the University of Helsinki, we took on the challenge and developed a novel challenging workflow for correlative light serial electron microscopy (Figure C).

Specifically, we labelled cilia using in-house designed viral vector (AAV8:GFAP-ARL13B-eGFP), which we delivered intracranially during postnatal Day 0-1. We then euthanised TwKO^{astro} at the age of contorted cilia phenotype manifestation (4-5 months), collected vibratome sections, introduced physical marks by applying a focused moving laser beam (branding), imaged regions of interest with confocal microscopy, completed fixation, staining, and embedding for serial electron microscopy, and finally performed serial block-face electron microscopy. We succeeded in imaging the region of interest, however the resulting resolution was not sufficient to resolve ciliary ultrastructure (Figure C). As already mentioned above, we acknowledge that applying newest FIB-SEM machines instead of block-face microscopy may improve the resolution, however this was not available to us.

After these significant but unfortunately inconclusive efforts in trying to image cilia from the brain parenchyma, we have decided not to continue with this line of experimentation for this manuscript. With further tool development and assay optimization from our end this is an area we would like to expand upon in the future.

Revision Plan

Workflow for correlative light scanning block-face electron microscopy

Figure C. Left: the workflow for correlative light scanning block-face scanning electron microscopy. Right: examples of branding marks (top panel) and micrographs from serial block-face scanning electron microscopy (lower panel, TwKO^{astro} brain cortex). Arrow points to a cilium. Boxed area marks cilia on several serial sections. Stars mark structurally abnormal mitochondria in an astrocyte of TwKO^{astro}. Arrowheads point to healthy mitochondria of neighboring cells.

Points raised by Reviewer #3:

1) One thing that would be important if possible is metabolomics. For example, is aspartate/asparagine levels altered that could trigger ISR. Also, whether L-2HG, fumarate or succinate levels ratio to aKG differ, as this ratio can modulate aKG dioxygenases like TETs. Metabolomics with the RNA seq would be helpful. But I realize it could be challenging from in vivo astrocytes.

Revision Plan

Au: We agree with the reviewer that if it would be possible to reliably measure metabolite levels from purified Twnk KO astrocytes, it would be very insightful. The gold standard in the field is to extract metabolites from intact tissues or cells using rapid techniques and controlled conditions. In contrast, the process of astrocyte purification takes about four hours, while half lives of many metabolites are much shorter. For example, in isolated mitochondria metabolites are converted in less than a minute (Labajova et al. 2006). Additionally, many metabolites are naturally labile, where the rate of degradation is dependent on many factors, including temperature. Astrocyte purification includes enzymatic digestion at +37 C, mechanical trituration, multiple centrifugations, and exposure to various reagents. Thus, the usefulness of metabolomics is highest from bulk tissues but not purified cells, as for the latter, quick stabilization of the metabolome to infer true levels of metabolites, to our opinion, is not currently possible. Furthermore, there are examples of metabolomics from astrocytes and neurons purified from mouse models with cell-specific knockouts of mitochondrial proteins, which fail to capture expected differences, possibly highlighting methodological limitations (Göbel et al. 2020; Motori et al. 2020). Spatial metabolomics, which could solve the issue, is still of limited accessibility.

Relevant to this point, we have previously published metabolomics analyses from cortical lysates of TwKO^{astro} mice (Ignatenko et al. 2020). Indeed, levels of some of the tricarboxylic acid cycle intermediates are changed in TwKO^{astro}, which could be one of the triggers of ISR. How these metabolite changes relate to ciliary changes and function in astrocytes will be an important area to focus in the future for our group. However, these are long term investigations beyond the scope of this study.

Reviewer #4 (Comments to the Authors (Required)):

The manuscript by Ignatenko et al shows that conditional knockout of the mitochondrial DNA helicase twinkle induces an multiciliogenesis program in astrocytes, which have a single primary cilium. This motile ciliogenesis program was activated by blocking EGFR signaling in cultured astrocytes, but not in heart-specific TwKO, reactive astrocytes, or aged astrocytes, demonstrating cell-type specificity. In TwKO^{astro} mice, astrocyte cilia are longer and misshapen, while ependymal multiciliated cells appeared superficial normal. This manuscript is extremely well written and a delight to read, the data - for the most part - are convincing, and this new and surprising connection between mitochondria and cilia will appeal to the broad readership of the JCB.

Au: We thank the reviewer for the favorable assessment of our study.

I agree with the assessments from Review Commons, and the Authors' proposed revision plan. It is critical to look at additional ciliary markers in both astrocytes and ependymal cells. For the latter, I do not think authors can conclude "Ependymal cilia do not show apparent abnormalities upon mitochondrial dysfunction" based on the immunofluorescent and SEM images shown in Figure 4B. To my eye, the cilia appear possibly misoriented in TwKO^{astro} animals. At a minimum, authors should use α -Acetylated tubulin for axonemes and α - γ tubulin for basal bodies (this antibody in conjunction with α -beta-catenin would also be useful for examining morphology and polarity of ependymal cilia basal bodies)

Revision Plan

Au: We thank the reviewer for the suggestions. As explained in our responses above, we have added considerable amount of new data for ciliary proteins, both by a sensitive proteomics approach using purified adult mouse cortical astrocytes, and by immunofluorescence (new Figs. 3D, 4G-H, S2F-G, and Suppl. tables 3 and 6). These data confirm that motile cilia proteins are indeed increased in parenchymal astrocytes, cell type that does not typically possess motile cilia.

We agree with the reviewer that our analysis of the ependymal cells is less detailed. To investigate motile ependymal cilia in detail, en-face dissection preparations of the ventricle would be needed with functional analysis of cilia motility and quantification of basal body orientation using TEM, ideally at different developmental stages (Guirao et al. 2010; Mirzadeh et al. 2010). In our manuscript focused on astrocytic primary cilia, we optimized the experimental conditions for the parenchymal astrocytes and used sagittal and coronal thin sections of the brain. Subventricular zone with ependymal cells is present on such sections, however these preparations are suboptimal for analysis of the orientation of lumen-facing ependymal cilia and do not allow to properly investigate ependymal cell polarity and directionality of cilia. Therefore, to study ependymal cells in detail is a separate study.

For our Short Report formatted manuscript, we prefer to keep the analysis of ependymal cells concise, and have now softened the conclusions about the morphology of ependymal cilia to a simple notion that ependymal cells of TwKO^{astro} are multiciliated and survive upon mitochondrial dysfunction (page 9, paragraph 2). This stand-alone finding does not change any of the major conclusions of the manuscript.

Cilia consume ATP yet have no mitochondria. There is a growing body of literature related to ciliary metabolic demands and energy sources, particularly in the kidney and kidney diseases. Nephron development is one of the upregulated pathways in TwKO^{astro} (Fig 2a). If discussion space were unlimited, it would be interesting to hear authors' thoughts on this potential connection to their work.

Au: We thank the reviewer for pointing out this very interesting potential links between ciliary energy metabolism and kidney disease, however, we would prefer to avoid including speculation in this direction into discussion of our manuscript because of space constraints with the Short Report format and the fact that our study is focused on the central nervous system. As for the energy metabolism of cilia, direct contact with mitochondria is not required to use the ATP generated by oxidative phosphorylation because the high energy carrier is transported out of the organelle into the cytosol.

This review on cilia in the brain was published while this manuscript was being revised. Tereshko et al write "In contrast to neurons, the role of cilia on mature astrocytes is poorly understood." This work by Ignatenko et al is important from fundamental cell biology and human disease (neurodegeneration and ciliopathies) perspectives.

Primary cilia in the postnatal brain: Subcellular compartments for organizing neuromodulatory signaling. Tereshko L, Turrigiano GG, Sengupta P. *Curr Opin Neurobiol.* 2022 Apr 8;74:102533. doi: 10.1016/j.conb.2022.102533. Online ahead of print.

Revision Plan

Au: We thank the reviewer for pointing out this study (which we now cite in our manuscript) and emphasizing the impact of our study; we full heartedly agree!

References in the responses.

- Avasthi, Prachee, and Wallace F. Marshall. 2012. "Stages of Ciliogenesis and Regulation of Ciliary Length." *Differentiation; Research in Biological Diversity* 83 (2): S30-42.
- Batiuk, Mykhailo Y., Filip de Vin, Sandra I. Duqué, Chen Li, Takashi Saito, Takaomi Saido, Mark Fiers, T. Grant Belgard, and Matthew G. Holt. 2017. "An Immunoaffinity-Based Method for Isolating Ultrapure Adult Astrocytes Based on ATP1B2 Targeting by the ACSA-2 Antibody." *The Journal of Biological Chemistry* 292 (21): 8874–91.
- Broekhuis, Joost R., Weng Y. Leong, and Gert Jansen. 2013. "Regulation of Cilium Length and Intraflagellar Transport." *International Review of Cell and Molecular Biology* 303: 101–38.
- Cahoy, John D., Ben Emery, Amit Kaushal, Lynette C. Foo, Jennifer L. Zamanian, Karen S. Christopherson, Yi Xing, et al. 2008. "A Transcriptome Database for Astrocytes, Neurons, and Oligodendrocytes: A New Resource for Understanding Brain Development and Function." *The Journal of Neuroscience: The Official Journal of the Society for Neuroscience* 28 (1): 264–78.
- Cho, Jung Hoon, Zipeng A. Li, Lifei Zhu, Brian D. Muegge, Henry F. Roseman, Toby Utterback, Louis G. Woodhams, Philip V. Bayly, and Jing W. Hughes. 2021. "Islet Primary Cilia Motility Controls Insulin Secretion." *BioRxiv*. <https://doi.org/10.1101/2021.12.14.472629>.
- Foo, Lynette C., Nicola J. Allen, Eric A. Bushong, P. Britten Ventura, Won-Suk Chung, Lu Zhou, John D. Cahoy, et al. 2011. "Development of a Method for the Purification and Culture of Rodent Astrocytes." *Neuron* 71 (5): 799–811.
- Forsström, Saara, Christopher B. Jackson, Christopher J. Carroll, Mervi Kuronen, Eija Pirinen, Swagat Pradhan, Anastasiia Marmyleva, et al. 2019. "Fibroblast Growth Factor 21 Drives Dynamics of Local and Systemic Stress Responses in Mitochondrial Myopathy with MtDNA Deletions." *Cell Metabolism* 30 (6): 1040-1054.e7.
- Gilroy, C., A. Singh, and E. Shahidi. 1995. "Cilia in the Porcine Bile Ductule: Motile or Sensory?" *Histology and Histopathology* 10 (2): 301–4.
- Gong, Shiaoqing, Chen Zheng, Martin L. Doughty, Kasia Losos, Nicholas Didkovsky, Uta B. Schambra, Norma J. Nowak, et al. 2003. "A Gene Expression Atlas of the Central Nervous System Based on Bacterial Artificial Chromosomes." *Nature* 425 (6961): 917–25.
- Guirao, Boris, Alice Meunier, Stéphane Mortaud, Andrea Aguilar, Jean-Marc Corsi, Laetitia Strehl, Yuki Hirota, et al. 2010. "Coupling between Hydrodynamic Forces and Planar Cell Polarity Orients Mammalian Motile Cilia." *Nature Cell Biology* 12 (4): 341–50.
- Göbel, Jana, Esther Engelhardt, Patric Pelzer, Vignesh Sakthivelu, Hannah M. Jahn, Milica Jevtic, Kat Folz-Donahue, et al. 2020. "Mitochondria-Endoplasmic Reticulum Contacts in Reactive Astrocytes Promote Vascular Remodeling." *Cell Metabolism* 31 (4): 791-808.e8.
- Ignatenko, Olesia, Dmitri Chilov, Ilse Paetau, Elena de Miguel, Christopher B. Jackson, Gabrielle Capin, Anders Paetau, Mugen Terzioglu, Liliya Euro, and Anu Suomalainen. 2018. "Loss of MtDNA Activates Astrocytes and Leads to Spongiotic Encephalopathy." *Nature Communications* 9 (1): 70.
- Ignatenko, Olesia, Joni Nikkanen, Alexander Kononov, Nicola Zamboni, Gulayse Ince-Dunn, and Anu Suomalainen. 2020. "Mitochondrial Spongiotic Brain Disease: Astrocytic Stress and Harmful Rapamycin and Ketosis Effect." *Life Science Alliance* 3 (9). <https://doi.org/10.26508/lsa.202000797>.

Revision Plan

- Ishikawa, Hiroaki, and Wallace F. Marshall. 2017. "Intraflagellar Transport and Ciliary Dynamics." *Cold Spring Harbor Perspectives in Biology* 9 (3). <https://doi.org/10.1101/cshperspect.a021998>.
- Jain, Raksha, Jiehong Pan, James A. Driscoll, Jeffrey W. Wisner, Tao Huang, Sean P. Gunsten, Yingjian You, and Steven L. Brody. 2010. "Temporal Relationship between Primary and Motile Ciliogenesis in Airway Epithelial Cells." *American Journal of Respiratory Cell and Molecular Biology* 43 (6): 731–39.
- Kasahara, Kyosuke, Ko Miyoshi, Shinki Murakami, Ikuko Miyazaki, and Masato Asanuma. 2014. "Visualization of Astrocytic Primary Cilia in the Mouse Brain by Immunofluorescent Analysis Using the Cilia Marker Arl13b." *Acta Medicinæ Okayama* 68 (6): 317–22.
- Khan, Shahzad S., Yuriko Sobu, Herschel S. Dhekne, Francesca Tonelli, Kerryn Berndsen, Dario R. Alessi, and Suzanne R. Pfeffer. 2021. "Pathogenic LRRK2 Control of Primary Cilia and Hedgehog Signaling in Neurons and Astrocytes of Mouse Brain." *ELife* 10 (October). <https://doi.org/10.7554/eLife.67900>.
- Kiesel, Petra, Gonzalo Alvarez Viar, Nikolay Tsoy, Riccardo Maraspini, Alf Honigmann, and Gaia Pigino. 2020. "The Molecular Structure of Primary Cilia Revealed by Cryo-Electron Tomography." *BioRxiv*. bioRxiv. <https://doi.org/10.1101/2020.03.20.000505>.
- Kremer, A., E. VAN Hamme, J. Bonnardel, P. Borghgraef, C. J. GuÉrin, M. Williams, and S. Lippens. 2020. "A Workflow for 3D-CLEM Investigating Liver Tissue." *Journal of Microscopy*, October. <https://doi.org/10.1111/jmi.12967>.
- Labajova, Anna, Alena Vojtiskova, Pavla Krivakova, Jiri Kofranek, Zdenek Drahota, and Josef Houstek. 2006. "Evaluation of Mitochondrial Membrane Potential Using a Computerized Device with a Tetraphenylphosphonium-Selective Electrode." *Analytical Biochemistry* 353 (1): 37–42.
- Liddelow, Shane A., Kevin A. Guttenplan, Laura E. Clarke, Frederick C. Bennett, Christopher J. Bohlen, Lucas Schirmer, Mariko L. Bennett, et al. 2017. "Neurotoxic Reactive Astrocytes Are Induced by Activated Microglia." *Nature* 541 (7638): 481–87.
- Luckner, Manja, Steffen Burgold, Severin Filser, Maximilian Scheungrab, Yilmaz Niyaz, Eric Hummel, Gerhard Wanner, and Jochen Herms. 2018. "Label-Free 3D-CLEM Using Endogenous Tissue Landmarks." *IScience* 6 (August): 92–101.
- Matsumoto, Mami, Masato Sawada, Diego García-González, Vicente Herranz-Pérez, Takashi Ogino, Huy Bang Nguyen, Truc Quynh Thai, et al. 2019. "Dynamic Changes in Ultrastructure of the Primary Cilium in Migrating Neuroblasts in the Postnatal Brain." *The Journal of Neuroscience: The Official Journal of the Society for Neuroscience* 39 (50): 9967–88.
- Mick, Eran, Denis V. Titov, Owen S. Skinner, Rohit Sharma, Alexis A. Jourdain, and Vamsi K. Mootha. 2020. "Distinct Mitochondrial Defects Trigger the Integrated Stress Response Depending on the Metabolic State of the Cell." *ELife* 9 (May). <https://doi.org/10.7554/eLife.49178>.
- Mirzadeh, Zaman, Fiona Doetsch, Kazunobu Sawamoto, Hynek Wichterle, and Arturo Alvarez-Buylla. 2010. "The Subventricular Zone En-Face: Wholemount Staining and Ependymal Flow." *Journal of Visualized Experiments: JoVE*, no. 39 (May). <https://doi.org/10.3791/1938>.
- Moser, Joanna J., Marvin J. Fritzler, and Jerome B. Rattner. 2009. "Primary Ciliogenesis Defects Are Associated with Human Astrocytoma/Glioblastoma Cells." *BMC Cancer* 9 (December): 448.
- Motori, E., I. Atanassov, S. M. V. Kochan, K. Folz-Donahue, V. Sakthivelu, P. Giavalisco, N. Toni, J. Puyal, and N-G Larsson. 2020. "Neuronal Metabolic Rewiring Promotes Resilience to Neurodegeneration Caused by Mitochondrial Dysfunction." *Science Advances* 6 (35): eaba8271.
- Odor, D. L., and R. J. Blandau. 1985. "Observations on the Solitary Cilium of Rabbit Oviductal Epithelium: Its Motility and Ultrastructure." *The American Journal of Anatomy* 174 (4): 437–53.

Revision Plan

- Sipos, Éva, Sámuel Komoly, and Péter Ács. 2018. "Quantitative Comparison of Primary Cilia Marker Expression and Length in the Mouse Brain." *Journal of Molecular Neuroscience: MN* 64 (3): 397–409.
- Sterpka, Ashley, and Xuanmao Chen. 2018. "Neuronal and Astrocytic Primary Cilia in the Mature Brain." *Pharmacological Research: The Official Journal of the Italian Pharmacological Society* 137 (November): 114–21.
- Sterpka, Ashley, Juan Yang, Matthew Strobel, Yuxin Zhou, Connor Pauplis, and Xuanmao Chen. 2020. "Diverged Morphology Changes of Astrocytic and Neuronal Primary Cilia under Reactive Insults." *Molecular Brain* 13 (1): 28.
- Sun, Shufeng, Rebecca L. Fisher, Samuel S. Bowser, Brian T. Pentecost, and Haixin Sui. 2019. "Three-Dimensional Architecture of Epithelial Primary Cilia." *Proceedings of the National Academy of Sciences of the United States of America* 116 (19): 9370–79.

September 26, 2022

RE: JCB Manuscript #202203019R

Prof. Anu Suomalainen-Wartiovaara
University of Helsinki
Stem Cells and Metabolism Research Program
Biomedicum Helsinki
Haartmaninkatu 8
Helsinki 00290
Finland

Dear Prof. Suomalainen-Wartiovaara:

Thank you for submitting your revised manuscript entitled "Mitochondrial dysfunction compromises ciliary homeostasis in astrocytes." We would be happy to publish your paper in JCB pending minor textual changes recommended by the reviewers as well as final revisions to necessary to meet our formatting guidelines (see details below).

A. MANUSCRIPT ORGANIZATION AND FORMATTING:

- 1) Text limits: Character count for Reports is < 20,000, not including spaces. Count includes title page, abstract, introduction, results, discussion, and acknowledgments. Count does not include materials and methods, figure legends, references, tables, or supplemental legends.
- 2) Figure formatting: Reports may have up to 5 main text figures. Scale bars must be present on all microscopy images, including inset magnifications. Molecular weight or nucleic acid size markers must be included on all gel electrophoresis. Please add scale bars to the inset magnification images in Figure S2D.
- 3) Statistical analysis: Error bars on graphic representations of numerical data must be clearly described in the figure legend. The number of independent data points (n) represented in a graph must be indicated in the legend. Statistical methods should be explained in full in the materials and methods. For figures presenting pooled data the statistical measure should be defined in the figure legends. Please also be sure to indicate the statistical tests used in each of your experiments (both in the figure legend itself and in a separate methods section) as well as the parameters of the test (for example, if you ran a t-test, please indicate if it was one- or two-sided, etc.). Also, if you used parametric tests, please indicate if the data distribution was tested for normality (and if so, how). If not, you must state something to the effect that "Data distribution was assumed to be normal but this was not formally tested."
- 4) Materials and methods: Should be comprehensive and not simply reference a previous publication for details on how an experiment was performed. Please provide full descriptions (at least in brief) in the text for readers who may not have access to referenced manuscripts. The text should not refer to methods "...as previously described."
- 5) For all cell lines, vectors, constructs/cDNAs, etc. - all genetic material: please include database / vendor ID (e.g., Addgene, ATCC, etc.) or if unavailable, please briefly describe their basic genetic features, even if described in other published work or gifted to you by other investigators (and provide references where appropriate). Please be sure to provide the sequences for all of your oligos: primers, si/shRNA, RNAi, gRNAs, etc. in the materials and methods. You must also indicate in the methods the source, species, and catalog numbers/vendor identifiers (where appropriate) for all of your antibodies, including secondary. If antibodies are not commercial please add a reference citation if possible.
- 6) Microscope image acquisition: The following information must be provided about the acquisition and processing of images:
 - a. Make and model of microscope
 - b. Type, magnification, and numerical aperture of the objective lenses
 - c. Temperature
 - d. Imaging medium
 - e. Fluorochromes
 - f. Camera make and model
 - g. Acquisition software

h. Any software used for image processing subsequent to data acquisition. Please include details and types of operations involved (e.g., type of deconvolution, 3D reconstitutions, surface or volume rendering, gamma adjustments, etc.).

7) References: There is no limit to the number of references cited in a manuscript. References should be cited parenthetically in the text by author and year of publication. Abbreviate the names of journals according to PubMed.

8) Supplemental materials: There are strict limits on the allowable amount of supplemental data. Reports may have up to 5 supplemental figures and 10 videos. Please also note that tables, like figures, should be provided as individual, editable files. A paragraph summary of all supplemental material should appear at the end of the Materials and methods section. Please include one brief sentence per item.

9) eTOC summary: A ~40-50 word summary that describes the context and significance of the findings for a general readership should be included on the title page. The statement should be written in the present tense and refer to the work in the third person. It should begin with "First author name(s) et al..." to match our preferred style.

10) Conflict of interest statement: JCB requires inclusion of a statement in the acknowledgements regarding competing financial interests. If no competing financial interests exist, please include the following statement: "The authors declare no competing financial interests." If competing interests are declared, please follow your statement of these competing interests with the following statement: "The authors declare no further competing financial interests."

11) A separate author contribution section is required following the Acknowledgments in all research manuscripts. All authors should be mentioned and designated by their first and middle initials and full surnames. We encourage use of the CRediT nomenclature (<https://casrai.org/credit/>).

12) ORCID IDs: ORCID IDs are unique identifiers allowing researchers to create a record of their various scholarly contributions in a single place. At resubmission of your final files, please consider providing an ORCID ID for as many contributing authors as possible.

B. FINAL FILES:

Thank you for this interesting contribution, we look forward to publishing your paper in Journal of Cell Biology.

Sincerely,

Richard Youle, PhD
Monitoring Editor
Journal of Cell Biology

Dan Simon, PhD
Scientific Editor
Journal of Cell Biology

Reviewer #2 (Comments to the Authors (Required)):

Thank you very much for addressing my concerns and clarifying points throughout the results and methods sections.

Reviewer #4 (Comments to the Authors (Required)):

Authors did a thorough, rigorous, and convincing job of addressing all of my concerns (and those of other reviewers). I appreciate the new discussion paragraph on cilia diversity and variations from 9+2 and 9+0 structures, which is not well appreciated in the field. I would encourage authors to add a brief paragraph on technical limitations of study, focusing on the extensive/heroic efforts they took to examine cilia in brain using immunofluorescence and CLEM. Cilia cell biology field has much to learn about cilia in the brain and the manuscript by Ignatenko et al is a major advancement/contribution. Exciting times ahead.

Authors may want to consider/cite this work in a technical limitations/future directions discussion

A serotonergic axon-cilium synapse drives nuclear signaling to alter chromatin accessibility.

Sheu SH, Upadhyayula S, Dupuy V, Pang S, Deng F, Wan J, Walpita D, Pasolli HA, Houser J, Sanchez-Martinez S, Brauchi SE, Banala S, Freeman M, Xu CS, Kirchhausen T, Hess HF, Lavis L, Li Y, Chaumont-Dubel S, Clapham DE. Cell. 2022 Sep 1;185(18):3390-3407.e18. doi: 10.1016/j.cell.2022.07.026. PMID: 36055200